# EhRacM differentially regulates macropinocytosis and motility in the enteric protozoan parasite *Entamoeba histolytica*

**Misato Shimoyama**[1]◉, **Kumiko Nakada-Tsukui**[1,2]◉, **Tomoyoshi Nozaki**[1]*

**1** Department of Biomedical Chemistry, Graduate School of Medicine, The University of Tokyo, Tokyo, Japan, **2** Department of Parasitology, National Institute of Infectious Diseases, Tokyo, Japan

◉ These authors contributed equally to this work.
* nozaki@m.u-tokyo.ac.jp

## Abstract

Macropinocytosis is an evolutionarily conserved endocytic process that plays a vital role in internalizing extracellular fluids and particles in cells. This non-selective endocytic pathway is crucial for various physiological functions such as nutrient uptake, sensing, signaling, antigen presentation, and cell migration. While macropinocytosis has been extensively studied in macrophages and cancer cells, the molecular mechanisms of macropinocytosis in pathogens are less understood. It has been known that *Entamoeba histolytica*, the causative agent of amebiasis, exploits macropinocytosis for survival and pathogenesis. Since macropinocytosis is initiated by actin polymerization, leading to the formation of membrane ruffles and the subsequent trapping of solutes in macropinosomes, actin cytoskeleton regulation is crucial. Thus, this study focuses on unraveling the role of well-conserved actin cytoskeleton regulators, Rho small GTPase family proteins, in macropinocytosis in *E. histolytica*. Through gene silencing of highly transcribed *Ehrho*/*Ehrac* genes and following flow cytometry analysis, we identified that silencing *EhracM* enhances dextran macropinocytosis and affects cellular migration persistence. Live imaging and interactome analysis unveiled the cytosolic and vesicular localization of EhRacM, along with its interaction with signaling and membrane traffic-related proteins, shedding light on EhRacM's multiple roles. Our findings provide insights into the specific regulatory mechanisms of macropinocytosis among endocytic pathways in *E. histolytica*, highlighting the significance of EhRacM in both macropinocytosis and cellular migration.

## Author summary

*Entamoeba histolytica* is an intestinal protozoan parasite that causes amoebic dysentery and liver abscesses in humans. This organism exploits macropinocytosis, a cellular process that engulfs extracellular fluids and particles, for its survival and pathogenicity. Although macropinocytosis is well-characterized in immune cells and cancer cells as it is essential for nutrient uptake, its mechanisms in pathogens, such as *E. histolytica*, remain less explored. Our research focused on the molecular mechanisms underpinning

**Data Availability Statement:** The RNAseq data discussed in this publication have been deposited in NCBI's Gene Expression Omnibus and are accessible through GEO Series accession number

GSE229171 (https://www.ncbi.nlm.nih.gov/geo/query/acc.cgi?acc=GSE229171). The mass spectrometry proteomics data have been deposited to the ProteomeXchange Consortium via the PRIDE partner repository with the dataset identifier PXD042282 (10.6019/PXD042282) and PXD051913 (10.6019/PXD051913) (For EhRacM and EhRacJ, respectively. Other relevant data are within the manuscript and its Supporting Information files.

**Funding:** This research was funded by Grants-in-Aid for Scientific Research (B) from the Japan Society for the Promotion of Science (JSPS) (JP18H02650 and JP21H02723 to T.N.), Fostering Joint International Research (B) from JSPS (JP19H03463 to K.N-T), Grant for Science and Technology Research Partnership for Sustainable Development (SATREPS) from Japan Agency for Medical Research and Development (AMED) and Japan International Cooperation Agency (JICA) (JP24jm0110022) to T.N., Grant for research on emerging and re-emerging infectious diseases from AMED (JP24fk0108680 to T.N., P23fk0108680j0401 and JP24fk0108683j0302 to K.N-T), NCGM International Research Fund from National Center for Global Health and Medicine (23A2017 to K.N-T), support from the University of Tokyo Pandemic Preparedness, Infection, and Advanced Research Center (UTOPIA) and AMED (JP243fa627001) to T.N., research grant from Ohyama Health Foundation to K.N-T, and research grant from JSPS for the world-leading innovative graduate study program for life science and technology (WINGS-LST) to M.S. The funders had no role in study design, data collection and analysis, decision to publish, or preparation of the manuscript.

**Competing interests:** The authors have declared that no competing interests exist.

macropinocytosis in this parasite, specifically examining the role of Rho small GTPase family proteins. These proteins are critical regulators of the actin cytoskeleton in eukaryotic cells. Our study reveals that one specific Rho small GTPase, EhRacM, is in the maturation of macropinosomes as well as in directing linear cell migration. The physiological significance of EhRacM in regulating both macropinocytosis and migration opens new avenues for understanding the role of Rho small GTPases in these signaling pathways, which could eventually lead to the development of new control measures against diseases caused by this parasite.

## Introduction

Macropinocytosis is an evolutionarily conserved mode of internalizing large amounts of extracellular fluid into a cell. It is one of the non-selective endocytic pathways by which any solute or small particles, such as cell fragments, viruses, or bacteria, can be internalized. Macropinocytosis serves in a range of physiological processes, including nutrient uptake, sensing, signaling, antigen presentation, and cell migration [1–4]. Macropinocytosis has been extensively studied in macrophages and cancer cells [5,6], but the role and molecular mechanisms of macropinocytosis in the pathogenesis of eukaryotic pathogens remain to be elucidated. Unlike other endocytic pathways, such as receptor-mediated endocytosis and phagocytosis, macropinocytosis is not initiated by the binding of ligands to cell surface receptors [4]. Instead, macropinocytosis is initiated by actin polymerization at the plasma membrane to generate extensions called membrane ruffles [1]. Ruffles can trap solutes into large (0.2–5 μm in diameter), irregular-shaped vesicles called macropinosomes [1]. After the internalization, the macropinocytic vesicles (macropinosomes) are acidified, matured, and subsequently neutralized before release through exocytosis [7–10].

*Entamoeba histolytica* is the enteric protozoan parasite and the causative agent of amebiasis, one of the most serious infectious diseases worldwide, with about 55,000 global deaths annually, particularly in low- and middle-income countries [11]. Endocytosis, including macropinocytosis, trogocytosis, and phagocytosis, is essential for survival, proliferation, and pathogenesis inside the host [3,12]. *E. histolytica* trophozoites are highly capable of macropinocytosis; they regularly internalize the volume of fluid, corresponding to ~8% of their cell volume in one hour [12]. While it has been known that actin rearrangement plays an important role in the initial stage of macropinocytosis in *E. histolytica* [13], very little is known about the molecular mechanisms of macropinocytosis in this parasite.

Rho/Rac small GTPase family (Rho hereinafter) proteins are highly conserved actin regulators. Besides actin cytoskeletal regulation in cell division and motility, Rhos are known to be involved in multiple mechanisms such as gene expression, vesicle trafficking, and endocytosis [14]. For instance, Rac1 was reported to be involved in macropinocytosis by regulation of membrane ruffling [15,16]. Rhos work as molecular switches by switching between GTP-bound active and GDP-bound inactive states, regulated by guanine-nucleotide exchange factors (GEFs), GTPase activating proteins (GAPs), and guanosine diphosphate dissociation inhibitors (GDI) [17]. Once activated, Rhos translocate to target membranes and interact with downstream effectors. The *E. histolytica* genome encodes 22 Rhos [18]; some have been investigated for their actin-related roles. For example, EhRho1 (EHI_029020) was reported to recruit two actin-binding proteins, EhFromin1 and EhProfilin1, during phagocytosis [19]. Overexpression of a constitutively active EhRacA caused a defect in erythrophagocytosis and surface receptor capping, the latter of which is believed to be vital for the evasion from the host

immune response [20]. Moreover, overexpression of a constitutively active EhRacG impaired cytokinesis, caused multinucleation of giant cells, and altered cell polarity of trophozoites due to the condensation of F-actin at one end of the cell [21].

In this report, in order to better understand how Rhos are involved in macropinocytosis through actin regulation in *E. histolytica*, we attempted to identify Rhos that are primarily, if not exclusively, involved in macropinocytosis. We first screened a panel of *E. histolytica* strains in which each gene of highly transcribed *Ehrho* genes was silenced by transcriptional gene silencing for the reduced macropinocytosis phenotype. Here, we demonstrate that EhRacM [for nomenclature, see Chung *et al.*, 2010 [18]] is involved in multiple processes, including macropinocytosis, migration, and signaling by reverse genetics, live imaging, and co-immunoprecipitation.

## Results

### Establishment of *E. histolytica rac* gene silenced strains

To identify and characterize Rhos (or Racs) that are involved in macropinocytosis in *E. histolytica* trophozoites, we established gene silenced strains for the ten most highly expressed *Ehrho/Ehrac* genes [22] using antisense small RNA-mediated transcriptional gene silencing [23,24]: EHI_070730 [(*EhracC* [18] or *Ehrho7* [25]], EHI_029020 (*Ehrho1B*) [18], XP_651936 [in the current database, EHI_013650 (*EhracR2*) and EHI_068240 (*EhracR1*), two identical copies present in two independent loci [18]], EHI_129750 [*EhracG* [18] or *Ehrho2* [25]], EHI_012240 [*EhracD1* [18] or *Ehrho5* [25]], EHI_197840 [*EhracA2* [18] or *Ehrho6* [25]], EHI_135450 [*EhracM* [18] or *Ehrho13* [25]], EHI_146180 [*EhracQ* [18] or *Ehrho16* [25]], EHI_194390 [*EhracJ* [18] or *Ehrho15* [25]], and EHI_192450 (*EhracP* [18] or *Ehrho14* [25])]. We examined the repression of the target gene expression by reverse transcriptase (RT)-PCR (Fig 1A) and quantitative real-time (qRT)-PCR (Fig 1B). Hereinafter, gene names underlined above are used in the current study. All strains, except for *EhracA2* and *EhracQ* gene silenced strains, showed reduced expression levels of corresponding genes compared to the mock strain. The XP_651936 (EHI_013650 (*EhracR2*) or EHI_068240 (*EhracR1*)) gene silenced strain was not obtained; it did not survive G418 selection despite multiple attempts, indicating that this gene is essential for survival. Of nine gene silenced strains, we tested if *EhracM* and *EhracJ* gene silenced strains showed any change in macropinocytosis. These gene silenced strains were chosen as they showed higher levels of gene silencing than other strains (Fig 1B). The gene-specific silencing of *EhracM* or *EhracJ* gene expression in corresponding gene silenced strains was confirmed by RNA-seq analysis (S6 and S7 Figs and S3–S8 Tables). EhRacM and EhRacJ contain a potential geranylgeranylation motif (CCXX) at the carboxyl terminus, which is generally found in Rab small GTPases [18]. One should note that most EhRhos have the carboxyl-terminal farnesylation motif (CXXX), which is commonly present in Rho small GTPases [18], suggesting the unique role of EhRacM and EhRacJ in membrane and cytoskeletal dynamics. However, these two possess typical Rho characters, including G boxes and Rho insert regions (S1 Fig) [26–28].

### *EhRacM* gene silencing caused an increase in macropinocytosis and cell volume

To investigate the potential involvement of EhRacM and EhRacJ in macropinocytosis, we measured macropinocytosis of *EhracM* and *EhracJ* gene silenced strains using flow cytometry (FACS). We incubated amoeba trophozoites in the medium containing Rhodamine B isothiocyanate (RITC)-dextran, and the amount of dextran uptake was estimated by measuring

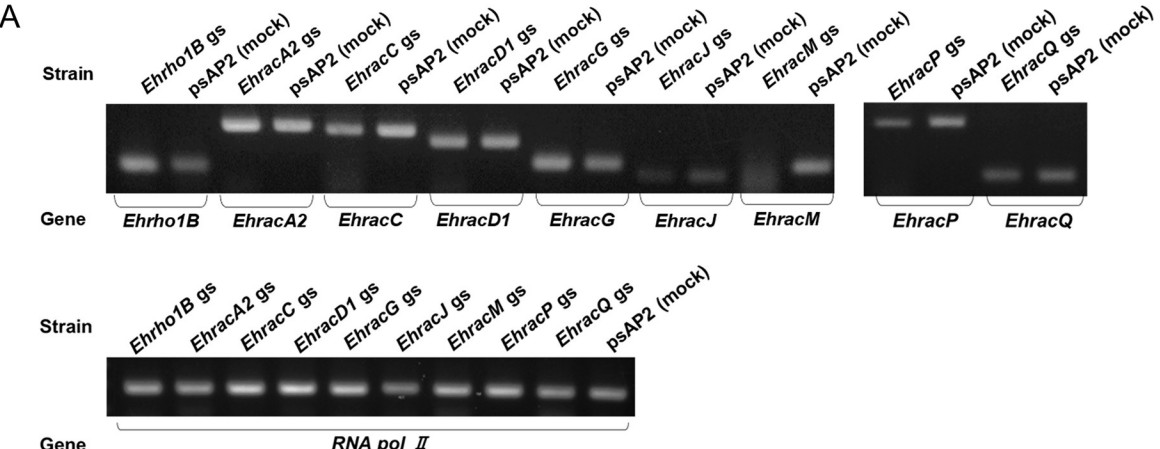

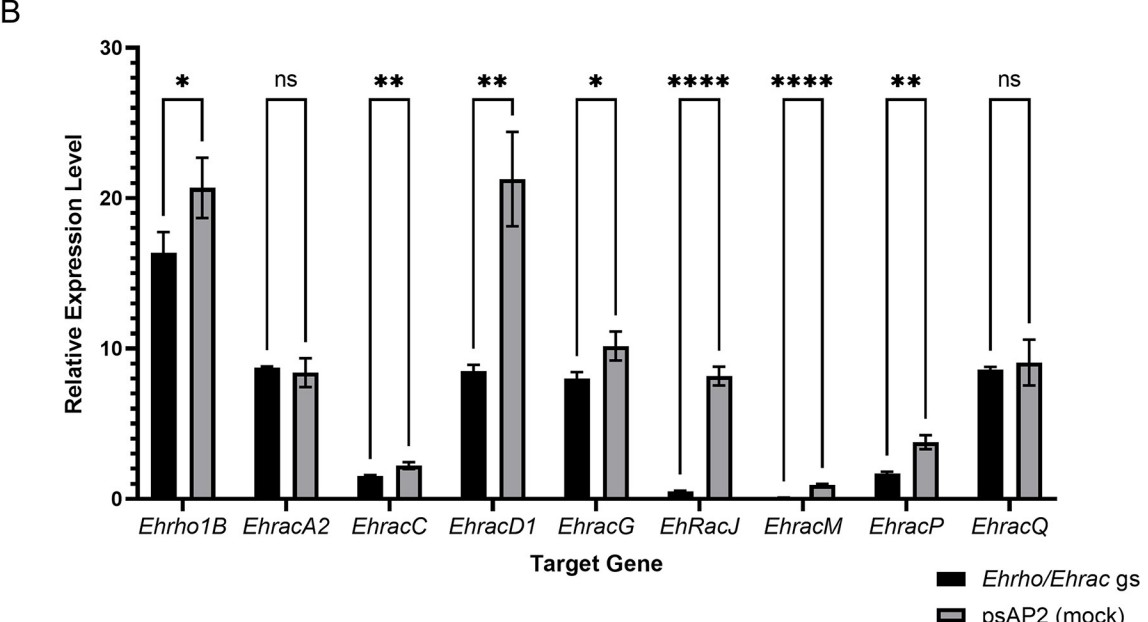

**Fig 1. Gene silencing of Rho small GTPases in *E. histolytica* trophozoites.** (A) Confirmation of gene silencing by RT-PCR analysis of *Ehrho/Ehrac* gene silenced (*Ehrho* gs) strains. Transcripts of indicated *Ehrho/Ehrac* and RNA polymerase II (*RNA pol II*, EHI_056690) genes were amplified by RT-PCR from cDNA isolated from the transformants and examined by agarose gel electrophoresis. Mock control generated by transfection with empty psAP2-Gunma vector was used as a reference (psAP2 (mock)). (B) qRT-PCR analysis of the relative expression level of the targeted *Ehrho/Ehrac* gene in each *Ehrho/Ehrac* gs strain and psAP2 mock strain. The steady state mRNA levels of *Ehrho/Ehrac* genes are indicated relative to that of *RNA pol II*. Statistical significance was examined with unpaired t-test, and p-values are shown in the graph (*p<0.05, **p<0.01, ***p<0.001, ****p<0.0001, ns: not significant). Error bars indicate standard deviations of three replicates.

fluorescence intensity. Gene silencing of *EhracM* significantly increased the amount of intra-cellular RITC-dextran, whereas *EhracJ* gene silencing did not affect it (Fig 2A). The increase in the RITC-dextran in trophozoites can be due to either an increase in the uptake or a decrease or a delay in the release of RITC dextran. To address this point, we monitored the decrease of RITC-dextran in the amoebas that had been preincubated with RITC-dextran, by chasing them in RITC-dextran free medium. However, no difference in the reduction rate of RITC-

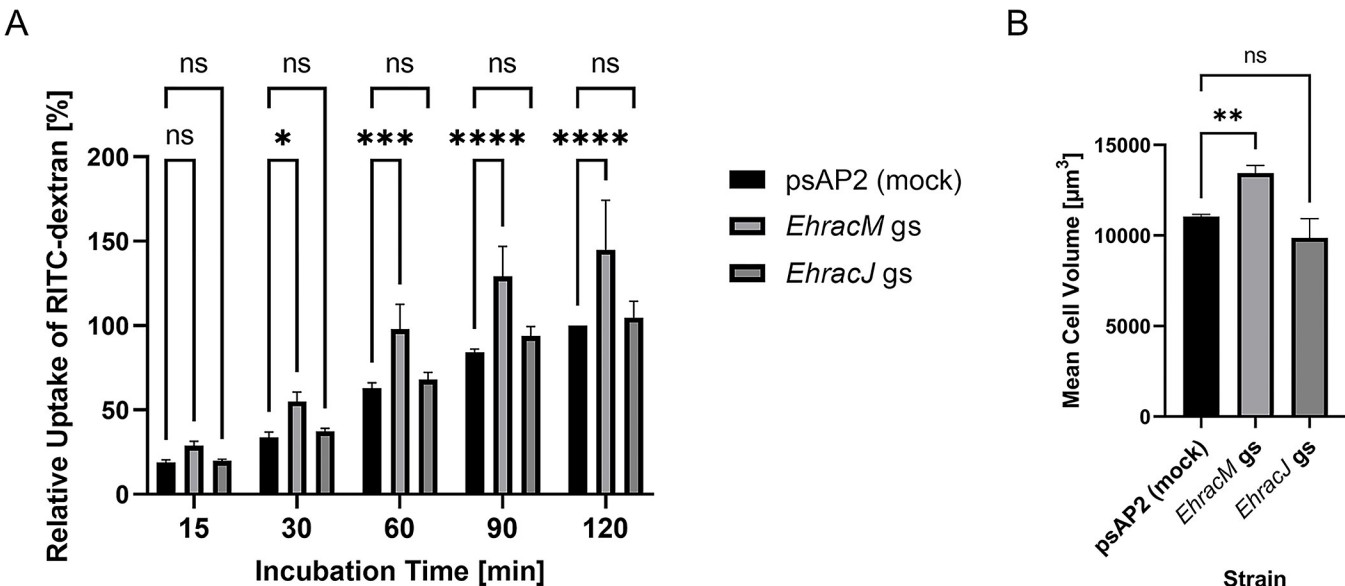

**Fig 2. *EhracM* gene silencing enhanced macropinocytosis.** (A) Macropinocytosis of *EhracM* and *EhracJ* gene silenced (gs) and psAP2 mock control strains. Trophozoites were incubated in RITC dextran-containing BIS medium to evaluate macropinocytosis. The fluorescence intensity of incorporated RITC-dextran by each strain was measured by FACS as described in Materials and Methods. RITC-dextran incorporation of each strain was estimated by calculating the geometric mean of the fluorescence intensity after the background signal from unlabeled parasites was subtracted from the fluorescence intensity in PE-A channel of each strain, and is shown relative to the value of mock strain at 120 min. Statistical significance was examined with Two-Way ANOVA (*$p < 0.05$, ***$p < 0.001$, ****$p < 0.0001$, ns: not significant). Error bars indicate standard deviations of three biological replicates. (B) Cell volume of *EhracM*, *EhracJ* gs, and control strains. Cell volume was estimated from the cell diameter measured by CASY system as described in Materials and Methods. Statistical significance was examined with One-Way ANOVA (**$p < 0.01$, ns: not significant). Error bars indicate standard deviations of three biological replicates.

dextran was observed between *EhracM* gene silenced and control strains (S2 Fig). Hence, *EhracM*-specific enhancement of macropinocytosis by its gene silencing indicates that EhRacM negatively and specifically regulates this process.

Besides macropinocytosis, *EhracM* gene silencing also significantly increased phagocytosis of 2-μm carboxylated beads (S3A Fig). However, phagocytosis of prelabeled and heat-killed Jurkat cells was not affected by *EhracM* gene silencing (S3B Fig). Given that the diameter of Jurkat cells is around 10-μm, which is larger than 2-μm carboxylated beads, we also measured phagocytosis of 10-μm carboxylated beads by *EhracM* gene silenced strain. Indeed, *EhracM* silencing only slightly increased the phagocytosis of 10-μm beads (S3D Fig). We also measured trogocytosis (nibbling or chewing of live target cells), by co-incubating amoeba trophozoites with prelabeled live Jurkat cells, followed by fluorescence measurement. We observed only a slight increase in trogocytosis by *EhracM* gene silenced strain (S3C Fig). These data indicate that EhRacM is selectively involved in the internalization of small (2 μm) artificial particles via phagocytosis and fluid phase markers via macropinocytosis but not involved in the phagocytosis of larger (10 μm) particles or dead cells (~10 μm).

Furthermore, *EhracM* gene silencing was also associated with the increase in cell volume (Fig 2B), which seems to be consistent with the increased macropinocytosis. However, *EhracM* silencing did not affect cell growth, as estimated with population doubling time (S4 Fig). This is a good contrast with the upregulation of macropinocytosis and enhanced proliferation often observed in cancer cells [29].

### *EhracM* silencing reduced directional persistence in cellular migration

Macropinocytosis is initiated by actin polymerization at the plasma membrane to generate extensions called membrane ruffles [1]. Since both macropinocytosis and cell migration are

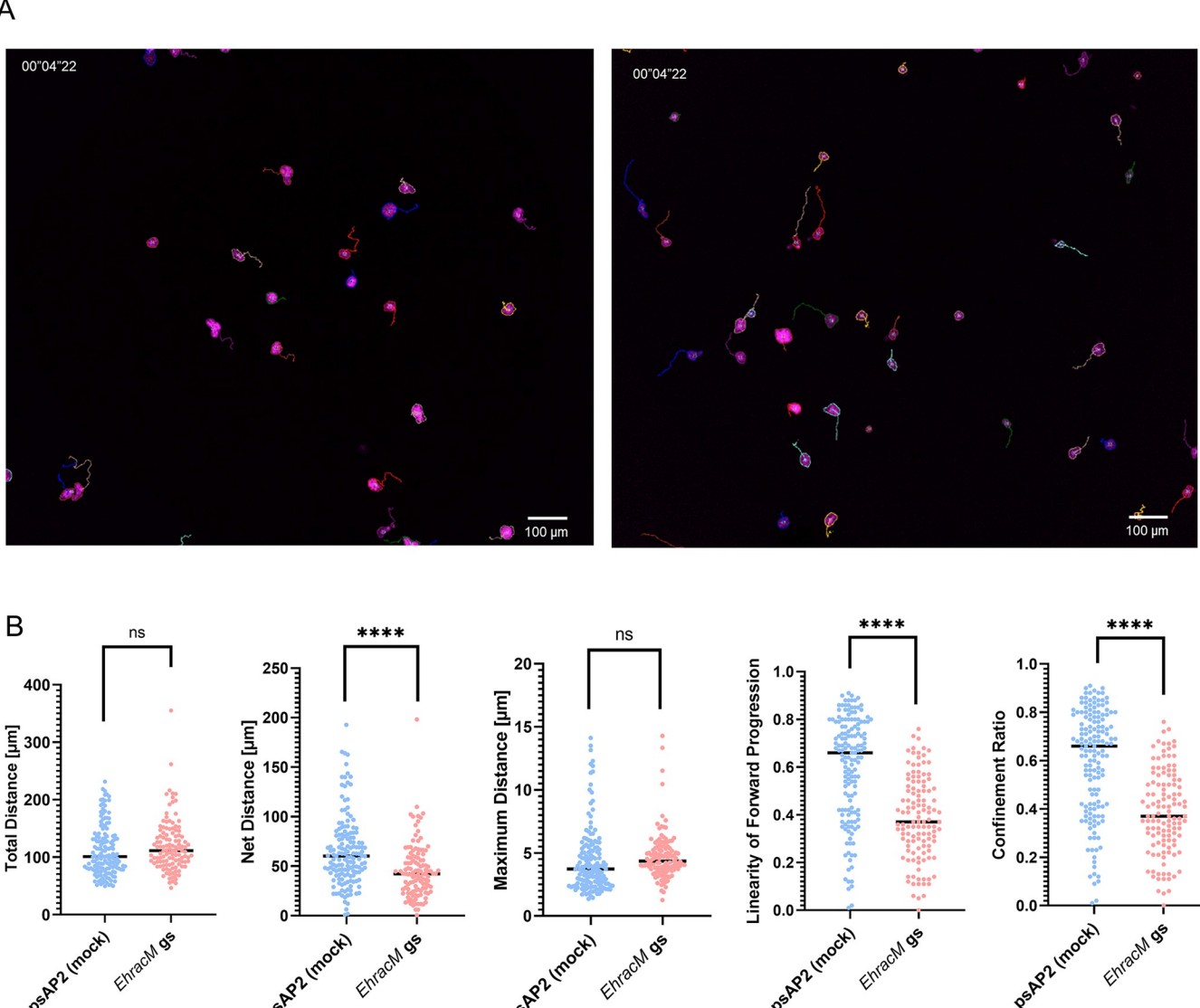

**Fig 3. Gene silencing of *EhracM* caused defects in linear and directed movement.** (A) Representative micrographs showing trajectories of live trophozoites (magenta) migrating on a glass surface. Left panel, *EhracM* gs strain; right panel, psAP2 mock control strain. The images were acquired with CQ1 (Yokogawa). Tracking of the cell trajectory was shown in lines. Bars, 100 μm. (B) A dot in five panels indicates the total distance, net distance, maximum distance, linearity of forward progression, and confinement ratio of each trophozoite during 300 seconds of incubation. 121 trophozoites of *EhracM* gs strain and 154 trophozoites of psAP2 mock strain were included in the analysis. Average values are shown as bars. Statistical significance was examined with an unpaired t-test (ns: not significant, ****p<0.0001).

actin-dependent, and possibly regulated by Rho small GTPases, we hypothesized that *EhracM* gene silencing may also affect cell migration either in a direct or an indirect way [30]. A link between macropinocytosis and cell migration has been demonstrated for other organisms, such as *Dictyostelium* and mammalian cells [31,32]. We examined cell motility of *EhracM* gene silenced and psAP2 mock strains. We measured the total distance, net distance, maximum distance, linearity of forward progression, and confinement ratio [34] (Fig 3A and 3B, and S1 Video). The net distance, linearity of forward progression, and confinement ratio decreased in *EhracM* gene silenced strain, whereas neither the total distance nor the maximum distance

was affected. These data indicate that linear and directed movement requires EhRacM, suggesting its role in cell motility, particularly in cell persistence [35].

## Cellular localization and dynamics of EhRacM in *E. histolytica* trophozoites

Typically, Rho small GTPases are primarily found in the cytosol when they are inactive, but once activated, they translocate to the plasma membrane and other loci to perform their functions. To investigate the subcellular localization of EhRacM in *E. histolytica* trophozoites, we generated transformants expressing EhRacM tagged with HA or GFP at the N-terminus (HA-EhRacM or GFP-EhRacM) [64,65]. Immunoblot analysis using anti-HA or anti-GFP antibodies confirmed the expression of HA-EhRacM and GFP-EhRacM in the transformant trophozoites. The expected molecular weights of the HA-EhRacM (22.4-kDa EhRacM plus three copies of 1.1-kDa HA tag, 25.7 kDa) and GFP-EhRacM (22.4-kDa EhRacM plus 26.9-kDa GFP tag, 49.3 kDa) were detected with anti-HA and anti-GFP antibodies as the predominant protein, along with some minor truncated bands seen in GFP-EhRacM-overexpressing trophozoites (Fig 4A and 4B).

Immunofluorescence assay (IFA) revealed that HA-EhRacM localized to the cytosol of trophozoites, was enriched on vesicles of ~5 μm diameter, and occasionally on the plasma membrane (Fig 4C). Live imaging analysis of GFP-EhRacM further confirmed their localization patterns (S9 Fig and S2 Video).

Subcellular fractionation followed by immunoblot assay further validated the localization pattern indicated by immunofluorescence and live imaging. Immunoblot analysis of cellular fractions of lysates from HA-EhRacM and GFP-EhRacM overexpressing transformants showed similar patterns. The 26- and 48.4-kDa bands were predominantly detected in the supernatant fractions (S100, which contains soluble cytoplasmic content) of lysates from HA-EhRacM and GFP-EhRacM transformants, respectively; smaller but significant proportions were also recovered in the low-speed pellet fraction at $13,000 \times G$ (P13, which contains the plasma membrane, nuclei, and large vacuoles) and the high-speed pellet at $100,000 \times G$ (P100, which contains small vesicles) (Fig 4D). The heavy subunit of Gal/GalNAc specific lectin, HgL [36], which contains a transmembrane region [37] and is known to be fractionated into P13 and P100, and CS1, a representative cytosolic protein and predicted to be exclusively found in S100, were used as controls. Although immunofluorescence and time-lapse live imaging very rarely showed the plasma membrane localization of EhRacM, a small but good amount of HA-EhRacM and GFP-EhRacM was detected in the heavy pellet fraction (P13) containing the plasma membrane.

## Overexpression of GFP-EhRacM decreased macropinocytosis

To examine the impact of overexpression of EhRacM on macropinocytosis, the macropinocytosis of RITC-dextran was assessed in GFP-EhRacM strain using FACS. Only amoebas expressing GFP-EhRacM were selected from the whole transformant population by gating on the green fluorescence channel, and RITC-dextran was measured on the red fluorescence channel. GFP-EhRacM-expressing amoebas showed a slight decrease in the uptake of RITC-dextran at all time points of ~120 min. Although the differences were not statistically significant, these data were consistent with the expected role of EhRacM in the negative regulation of macropinocytosis (S10 Fig).

## GFP-EhRacM was not recruited at the macropinocytic cup at the initial step but enriched on macropinosomes at subsequent maturation steps

To further investigate the role of EhRacM in macropinocytosis, live imaging of RITC-dextran macropinocytosis was conducted to gain insights into how EhRacM is involved in this process.

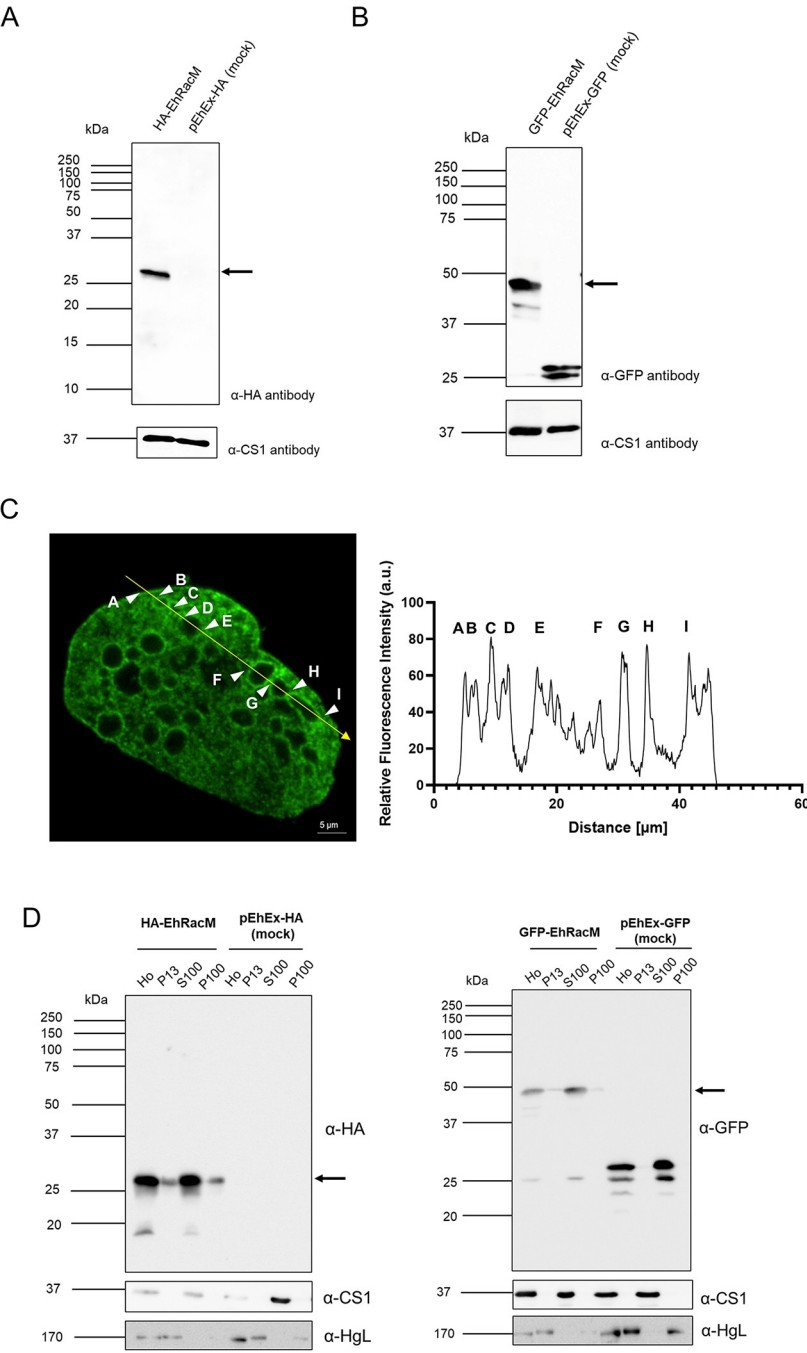

**Fig 4. Expression and cellular localization of HA—and GFP-fused EhRacM in *E. histolytica* trophozoite.** (A and B) Immunoblot detection of HA-EhRacM and GFP-EhRacM in *E. histolytica* transformants. Approximately 30 μg of total lysates from HA- or GFP-fused EhRacM expressing transformants and mock-transfected control (pEhEx-HA or pEhEx-GFP) were subjected to SDS-PAGE and immunoblot analysis using anti-HA monoclonal antibody (A), anti-GFP monoclonal antibody (B), and anti-CS1 polyclonal antibody (loading control). An arrow indicates HA-EhRacM (A) or GFP-EhRacM (B). (C) The immunofluorescence image of a representative trophozoite of HA-EhRacM strain. HA-EhRacM was visualized with anti-HA antibody (left). A yellow arrow indicates the trajectory of the fluorescence intensity plot shown in the right panel. The arrowheads with alphabets indicate ~10 positions where HA-EhRacM intensity peaked. (D) Immunoblot analysis of HA-EhRacM (left panel) and GFP-EhRacM (right panel) in fractionated cell lysates. Homogenate (Ho) from trophozoites of HA-EhRacM/GFP-EhRacM expressing transformant was fractionated by centrifugation into the low-speed pellet (p13, the pellet fraction of 13,000 × G centrifugation), the high-speed pellet (p100, the pellet fraction of 100,000 × G centrifugation) and the supernatant fractions (s100, the supernatant fraction of 100,000 × G centrifugation). These fractions were subjected to immunoblot analysis using anti-

HA/anti-GFP, anti-CS1 (a cytosolic protein), and anti-HgL (a membrane protein) antibodies. The arrows indicate the approximate sizes of HA-EhRacM (left) and GFP-EhRacM (right), respectively.

Given that Rho small GTPases are well-known actin cytoskeleton regulators, and actin recruitment has been demonstrated during the initial stage of macropinocytosis in *E. histolytica* [13], EhRacM was assumed to be recruited to the macropinocytic cup. However, contrary to the expectation, GFP-EhRacM was not enriched in the macropinocytic cup. Instead, GFP-EhRacM was gradually recruited to and began to decorate dextran-containing macropinosomes within 1–3 minutes (~42 sec in Fig 5 and S3 Video).

To determine whether and when GFP-EhRacM is disassociated from macropinosomes, amoebas were incubated with the medium containing RITC-dextran for 15 minutes and then further incubated in a dextran-free medium for 2 hours. GFP-EhRacM remained associated with macropinosomes for up to 2 hours (Fig 6). These results suggest that EhRacM is not involved in the early step of macropinocytosis, but in the maturation process after the closure of macropinosomes.

## GFP-EhRacM was recruited to macropinosomes after the removal of F-actin envelope

In order to better understand the timing of EhRacM recruitment to a nascent macropinosome, we examined the localization of GFP-EhRacM and F-actin stained by phalloidin. It has been previously reported that F-actin is recruited to the macropinocytic cup, surrounds the macropinosome for a minute, and is dissociated from the macropinosome [13]. As demonstrated above, GFP-EhRacM was not enriched on the macropinocytic cups (Fig 7, upper panels). In addition, most of the nascent macropinosomes decorated with F-actin envelope were not colocalized with GFP-EhRacM (Fig 7, lower panels). Instead, we observed the decoration of macropinosomes started ~40 seconds after the formation of the macropinocytic cup (Fig 5 and S3 Video). Thus, GFP-EhRacM seems to be recruited to macropinosomes after the removal of the F-actin envelope.

In order to further confirm this observation, we observed the dynamism of GFP-EhRacM together with SiR-Actin in the course of RITC-dextran macropinocytosis. F-actin staining by SiR-Actin was validated by the double staining of F-actin using SiR-Actin and phalloidin in fixed trophozoites [38] (S11 Fig). Here, we confirmed that GFP-EhRacM is recruited to the macropinosomes as the F-actin envelope is removed since the GFP-EhRacM's fluorescence intensity at the edge of macropinosome increased from 30 to over 40 (Figs 8 and S12, and S4 Video).

## Involvement EhRacM in vesicular trafficking and the ubiquitin/proteasome pathway was inferred by interactome analysis

As the above imaging and FACS analyses suggested the role of EhRacM in vesicular trafficking and endosome maturation, we investigated the interactome of EhRacM by co-immunoprecipitation (co-IP) and mass spectrometry to identify binding proteins of EhRacM. Fourteen proteins showed a quantitative value (QV) of >2 fold higher in the co-IP sample obtained from HA-EhRacM-expressing strain compared to that from HA mock control strain throughout three independent biological replicates, with their average QV being at least >1 (Table 1). Of those candidate proteins, Rap/Ran GAP (EHI_197320), DH domain-containing protein (Rho-GEF) (EHI_158230), and uridine/cytidine kinase (EHI_196580) were exclusively identified as binding proteins of HA-EhRacM. Apart from them, we identified Ras-related proteins [Ras

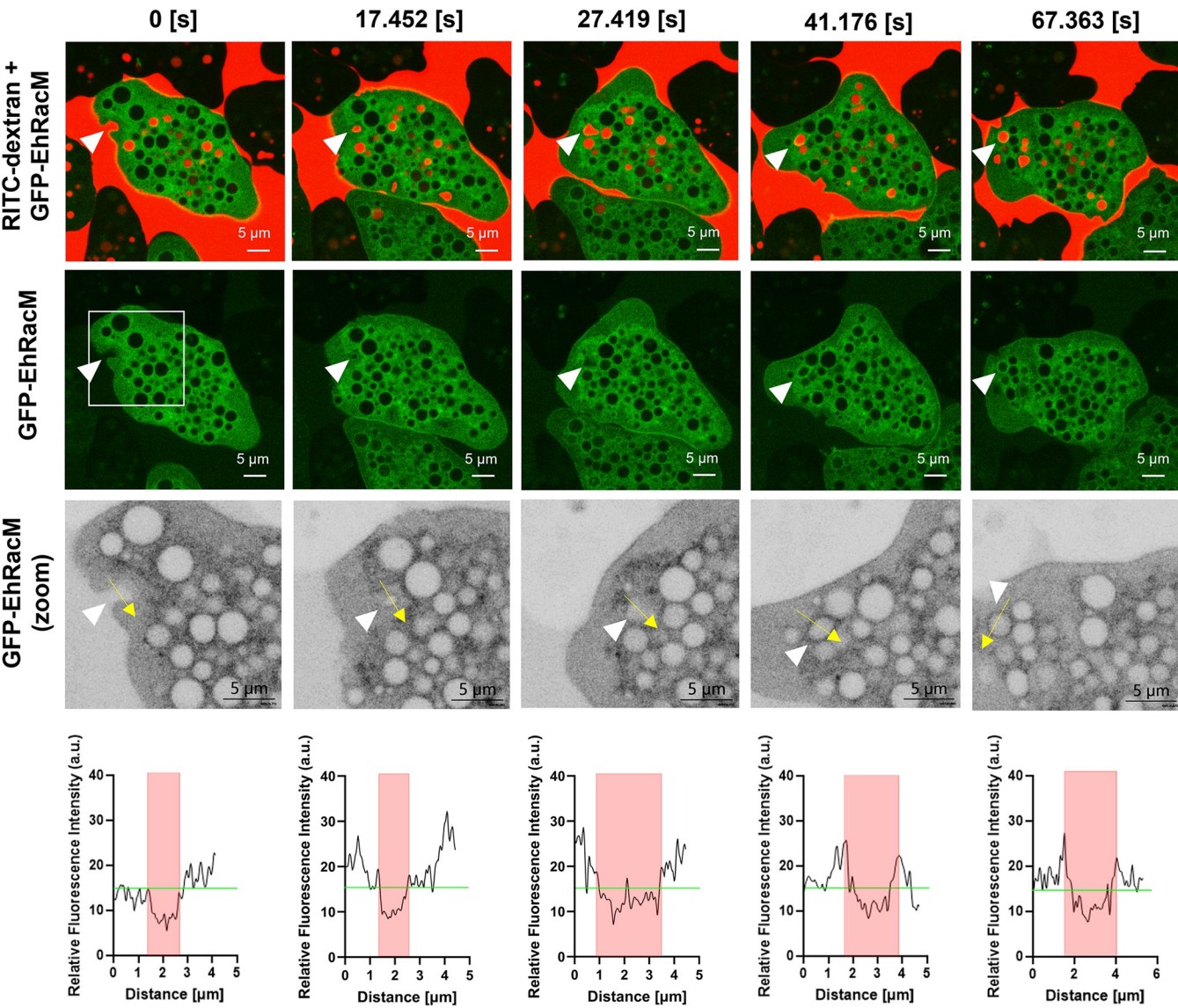

**Fig 5. GFP-EhRacM was recruited to the dextran-containing macropinosomes.** Montage of live imaging of GFP-EhRacM (green) expressing trophozoites that were incubated with RITC dextran (red). The white arrowheads show the site of macropinocytic cup formation and the resultant macropinosome. Three times-magnified images of the regions enclosed by a white square are displayed in the third row in an inverted grayscale. The fourth row shows the fluorescence intensity plots along with the trajectory of yellow arrows depicted in the corresponding third-row panels. Macropinosome areas are highlighted in red, with the GFP signal intensity at the initial macropinocytic cup formation (approximately 18) shown in green. Bars, 5 μm.

family GTPase (EHI_004860) and Ras GEF (EHI_079250)], Rab-related proteins [Rab-GAP TBC domain-containing protein (EHI_069260), Rab family GTPase (EhRab1B) (EHI_146510)], and ubiquitin-related proteins [RING-type domain-containing protein (EHI_013240), E3 ubiquitin-protein ligase listerin (EHI_190430), and proteasome regulatory subunit (EHI_005870)].

Subsequently, in order to predict the connection among potential hit proteins. We analyzed the proteins using the STRING protein-protein interaction network (S13B Fig). We included 107 proteins as hit proteins, which showed higher QV in HA-EhRacM than mock throughout all three trials, with their average QV being at least >1 (S9 Table and S13A Fig). In this analysis, we found that proteins related to proteasome, ribosome, and pyrimidine metabolism were

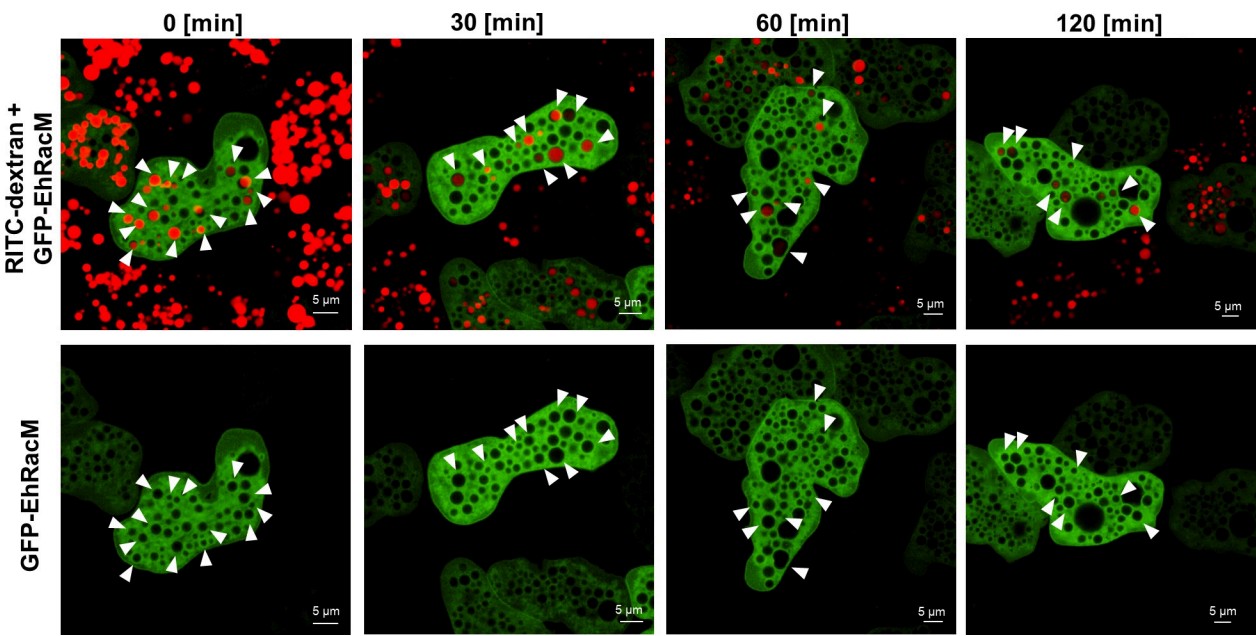

**Fig 6. GFP-EhRacM persisted on macropinosomes even after 2 hours.** Montage of live imaging of GFP-EhRacM expressing trophozoites incubated with RITC dextran for 15 minutes and then chased in a dextran-free BIS medium for the indicated time. White arrowheads show the macropinosomes decorated by GFP-EhRacM. Bars, 5 μm.

notably identified as hit proteins. Moreover, we conducted gene ontology (GO) enrichment analysis for these hit proteins (Fig 9). Here, we also found the enrichment of proteins related to proteasome (Fig 9A). The classification by molecular function indicated that proteins related to proteasome-activating activity were remarkably enriched (Fig 9B).

We also conducted interactome analysis with HA-EhRacJ and revealed that EhRacM and EhRacJ had completely different binding partners (S10 Table and S14 Fig). Instead of proteasome-related or ribosome-related proteins, proteins related to the actin filament capping were remarkably enriched in EhRacJ expressing strain (S14A Fig). Moreover, in terms of molecular function, proteins related to phosphatidylinositol bisphosphate binding or actin-binding were notably identified (S14B Fig). Actually, among the EhRacJ binding proteins, we specifically identified 18 cytoskeleton binding and regulator proteins exclusively in the HA-EhRacJ sample with QV exceeding 5, and 27 such proteins showed over 3-fold higher QV than mock control (S10 Table). These findings support the idea that EhRacJ is a typical actin-regulating Rho, whereas EhRacM has an atypical role in cytoskeleton regulation.

## Discussion

### Identification of EhRacM as a negative regulator of macropinocytosis

We screened ten most highly expressed *Ehrho* genes out of 22 [18] to identify *Ehrho/Ehrac* genes that are involved in macropinocytosis by generating gene silenced strains (Fig 1). By this screening, we successfully identified EhRacM as a negative regulator of macropinocytosis (Fig 2A). To date, several studies have demonstrated specific roles of *E. histolytica* Rho proteins in endocytic pathways, such as EhRacA [20], EhRho1 [19], EhRacG [21], and EhRho5 (EhRacD1 [18]) [39]. This study is the first to identify EhRacM as a macropinocytosis-specific regulator, independent from other endocytic pathways.

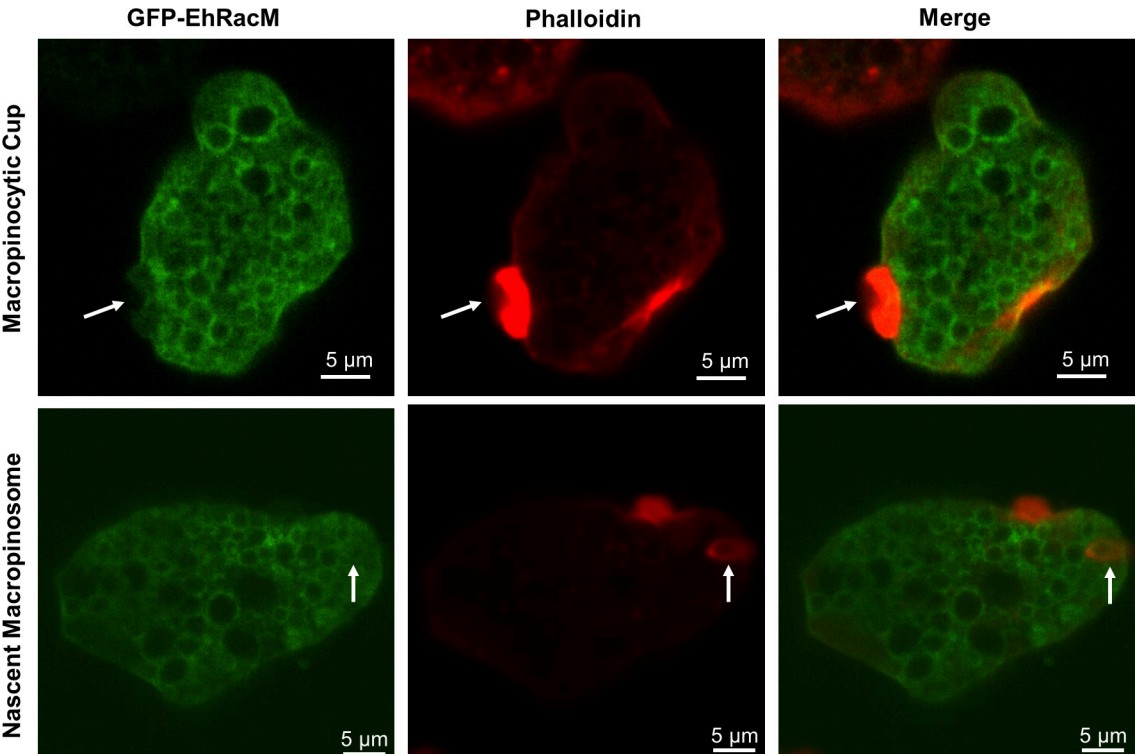

**Fig 7. GFP-EhRacM was not recruited to actin-rich macropinocytic cups nor nascent macropinosomes.** Representative images of GFP-EhRacM expressing trophozoites. F-actin was visualized by phalloidin. The white arrows depict the macropinocytic cup (in the upper row) and a semi-closed macropinosome with an F-actin envelope being formed (in the lower row). Note that GFP-EhRacM is not associated with either the macropinocytic cup or the enclosing macropinosome. Bars, 5 μm.

Although the mechanisms of EhRacM-mediated macropinocytosis regulation remain elusive, previous findings in other organisms suggest an involvement of the Ras signaling pathway (Tables 1 and S9). Ras has been known to principally regulate macropinocytosis by activating phosphatidylinositol 3-kinases (PI3Ks) in various organisms and cell types, including mammalian fibroblasts and the social amoeba *Dictyostelium* [40–43]. In good agreement, our co-IP data supports potential interactions between EhRacM and Ras and Ras-related proteins. It is plausible that EhRacM sequesters these proteins in the cytosol and prevents their translocation to the plasma membrane. Increased RasGAP (EHI_096700) and decreased Ras family GTPase (EHI_074750) transcripts in *EhracM* gene silenced strain (S3 and S4 Tables) also suggest a negative feedback mechanism in response to the altered signaling dynamics.

## EhRacM positively regulates cell motility and signaling pathways

Our study also indicates that EhRacM also regulates directional cell migration. It has been shown that Rac1 and Cdc42 influence directional persistence in cell migration by stabilizing lamellipodia [44,45]. However, no interaction between EhRacM and cytoskeleton regulation has been demonstrated in our study. Instead, EhRacM may regulate cell motility by modulating signaling pathways. It was previously shown in hematopoietic cells that RhoH is not directly involved in cytoskeletal regulation by actin regulation, but indirectly predominantly involved in cell proliferation and survival via signaling [46].

It is well established in mammalian cells and *Dictyostelium* that directional cell migration, e.g., chemotaxis, is regulated by receptor-mediated signaling involving Ras, PI3K, Rho, and the

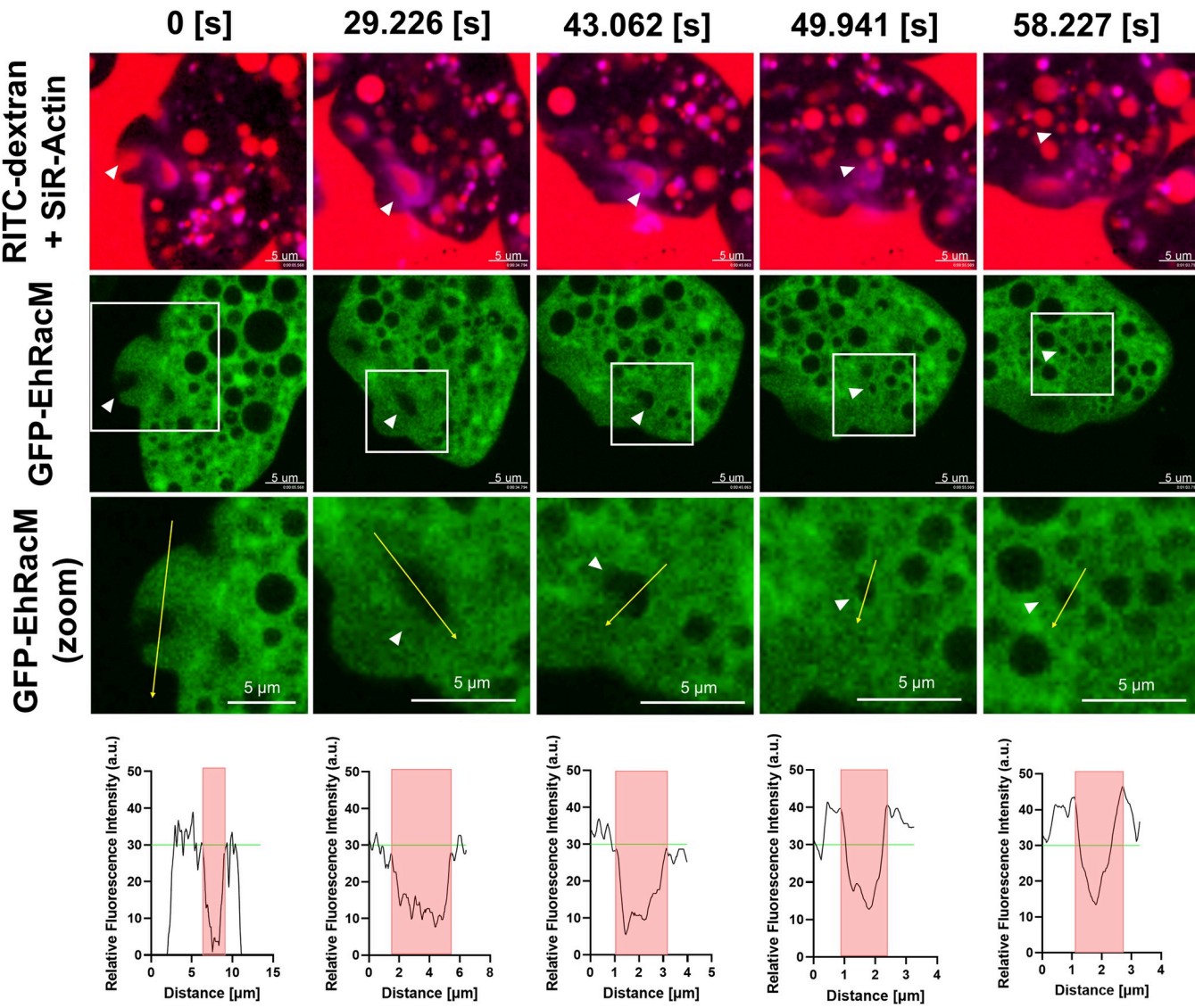

**Fig 8. GFP-EhRacM was recruited to macropinosomes following the removal of the F-actin envelope.** Montage of live imaging time series of a representative GFP-EhRacM expressing trophozoite in which macropinocytosis was monitored. F-actin was visualized using SiR-Actin (magenta), while GFP-EhRacM expressing trophozoites were shown in green (second row). The third-row panels are expanded images of the enclosed area by the white squares in the second-row panels. The trophozoites were incubated with RITC dextran (red). The white arrowheads indicate the site of macropinosome formation and the resultant macropinosome. The fourth-row panels show the GFP-EhRacM's fluorescence intensity plot along with the trajectory of yellow arrows depicted in the corresponding third-row panels. Macropinosome areas are highlighted in red, and the mean GFP-EhRacM's signal intensity at the initial macropinocytic cup (approximately 30) is shown in green. Bars, 5 μm. The F-actin envelope dissociation was captured at 49.941 [s].

target of rapamycin (TOR) [47,48]. In *E. histolytica*, PI3Ks have also been identified as key regulators of directional sensing and the sustained extension of pseudopods in various forms of taxis [49–52]. These insights suggest a complex interplay of signaling pathways that orchestrate the regulation of the pseudopod, a critical structure for directional movement in this parasite. Consequently, disruptions in these signaling pathways potentially caused by *EhracM* gene silencing could significantly affect amoebic migration mechanisms. In fact, we observed down/upregulation of gene expression of Ras small GTPase [EHI_074750 (down)], RasGEF [EHI_197190 (down), EHI_096700 (up)] and RhoGEF [EHI_150410 (down), EHI_152020 (down), EHI_068050 (down), EHI_110980 (up)], RhoGAP [EHI_072080 (down)], and Rho

**Table 1. List of hits in HA-EhRacM co-IP.**

| Accession Number | Description | Mean QV of mock | Mean QV of HA-EhRacM | QV Ratio (HA-EhRacM/mock) |
|---|---|---|---|---|
| EHI_197320 | Uncharacterized protein (Rap GEF) | 0 | 17.55 | - |
| EHI_158230 | DH domain-containing protein | 0 | 16.16 | - |
| EHI_196580 | Uridine/cytidine kinase | 0 | 4.96 | - |
| EHI_135450 | Rho family GTPase (EhRacM) | 1.76 | 131.94 | 75.03 |
| EHI_004860 | Ras family GTPase | 0.21 | 4.36 | 20.51 |
| EHI_112850 | DUF2807 domain-containing protein | 0.21 | 4.35 | 20.45 |
| EHI_069260 | Rab-GAP TBC domain-containing protein | 0.21 | 3.39 | 15.94 |
| EHI_177500 | DUF5857 domain-containing protein | 0.28 | 2.39 | 8.43 |
| EHI_079250 | Ras GEF | 0.88 | 3.61 | 4.11 |
| EHI_051730 | Uridine/cytidine kinase | 2.30 | 8.04 | 3.45 |
| EHI_013240 | RING-type domain-containing protein | 1.76 | 5.63 | 3.20 |
| EHI_190430 | E3 ubiquitin-protein ligase listerin | 2.09 | 6.17 | 2.95 |
| EHI_005870 | Proteasome regulatory subunit, putative | 1.67 | 4.52 | 2.71 |
| EHI_146510 | Rab family GTPase (EhRab1B) | 2.52 | 6.71 | 2.67 |
| EHI_178490 | Isopentenyl phosphate kinase | 1 | 2.65 | 2.65 |

Hits in three independent co-IPs, which showed more than two-fold higher quantitative value (QV) in HA-EhRacM than in mock strain in all independent analyses, are listed (Only the mean values of QV from three experiments are shown). Hits exclusively identified in HA-EhRacM are highlighted by a shadow. Hits are listed in order of QV ratio.

small GTPase [EHI_067220 (down)] under *EhracM* gene silencing (S3 and S4 Table). In addition, GO enrichment analysis suggests the downregulation of Ras- or Rho-related genes (S5 Fig). The involvement of EhRacM in the Ras-mediated signaling pathway was also suggested by the findings of Ras small GTPase (EHI_004860), RasGEF (EHI_079250), RasGAP (EHI_001910), Rho small GTPases (EHI_146180, EHI_070730, and EHI_181250), and Rho GDI (EHI_185440 and EHI_147570) in the co-IP of HA-EhRacM (S9 Table).

It is also possible that enhanced macropinocytosis could reduce motility as both macropinocytosis and cell motility depend on actin-mediated cytoskeletal rearrangements, and the resources for actin-related components are always limited [30]. EhRacM may have a balancing function between macropinocytosis and directional migration, similar to EhRho5 [EhRacD1 [18]], which positively regulates cytoskeletal reorganization toward both macropinocytosis and pseudopod formation [39].

## Unique characteristics of EhRacM

Although we mainly characterized EhRacM in this study, EhRacJ was also well repressed by gene silencing and used as a good comparator in this study. EhRacJ and EhRacM showed clear differences in their subcellular localization (Fig 4 and S8 Fig), but exhibited 46% identity at the amino acid sequence level, sharing four conserved G box GDP/GTP-binding motifs [53]. Despite this similarity, they are placed in distinct clades on the phylogenetic tree [18]. Alpha-Fold2-based predictions of their 3D structures [54,55] reveal that EhRacJ possesses an extended intrinsically disordered region, which is absent in EhRacM (S15 Fig). This structural variance might contribute to their different localizations; EhRacM is associated with the surface of macropinosomes, whereas EhRacJ is found on the plasma membrane.

The difference in function is further highlighted by their interactome analysis. HA-EhRacJ interactome showed significant enrichment of proteins related to cytoskeletal regulation (S14 Fig), supporting its role as a typical Rho family actin cytoskeleton regulator. In contrast,

A

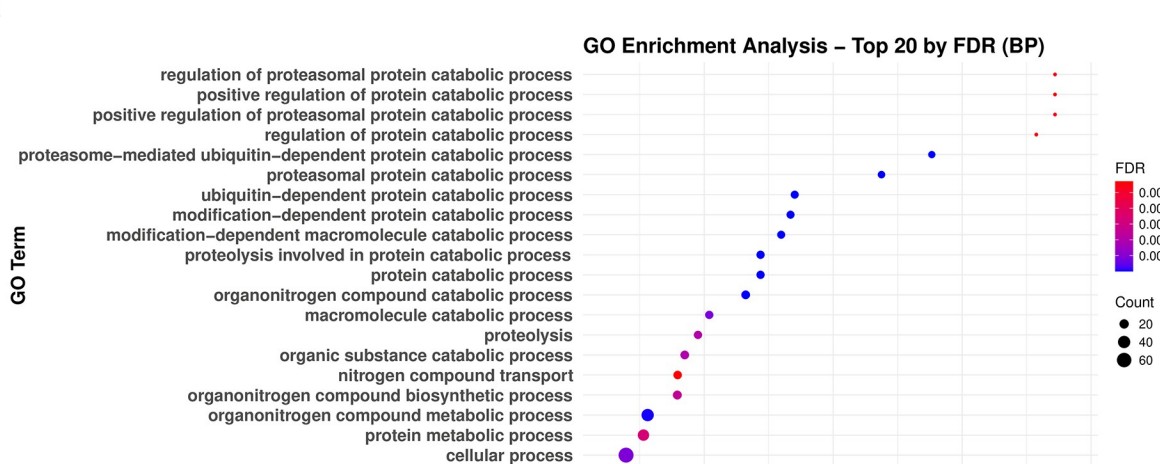

B

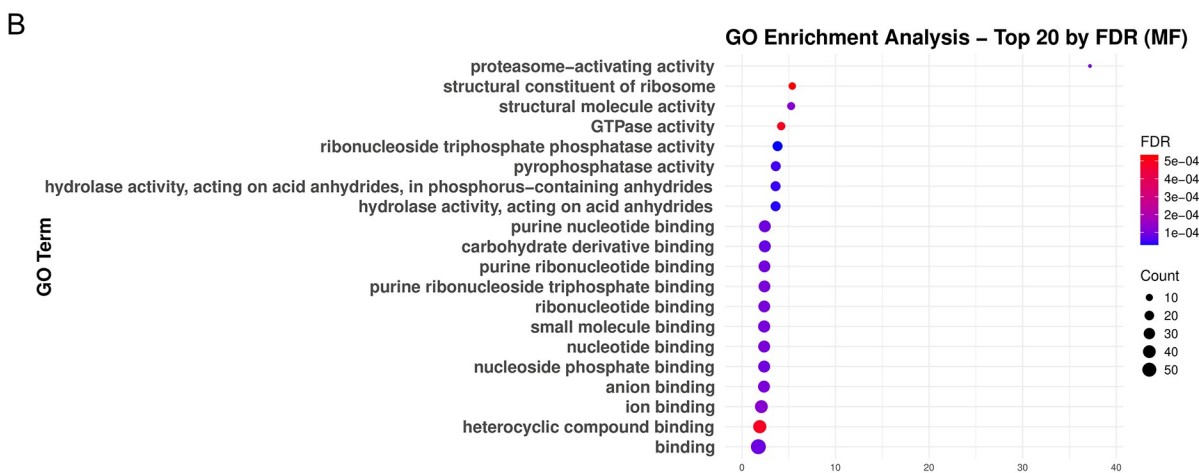

**Fig 9. GO enrichment analysis of EhRacM binding protein candidates identified by HA-EhRacM co-IP.** The results of PANTHER GO enrichment analysis on 107 hit proteins obtained from HA-EhRacM co-IP. Proteins were classified by biological process (BP) (A) and molecular function (MF) (B). Twenty GO terms were selected in descending order of fold enrichment for each entry. Each dot size reflects the count size, whereas its color reflects the FDR. The x-axis indicates fold change.

EhRacM interacts with proteins involved in signal transduction, including those associated with Ras, ribosome, proteasome, and pyrimidine metabolism (Table 1, S9 Table, Fig 9, and S13 Fig). The identification of the E3 ubiquitin ligase listerin, ubiquitin-related proteins, and proteasome-subunits (Table 1) may suggest that EhRacM is involved in ribosome-associated protein quality control [56,57]. Interestingly, we also discovered pyrimidine metabolism-related proteins exclusively in the EhRacM interactome (EHI_196589 and EHI_051730). In other organisms, uridine-cytidine kinase is recognized as a limiting enzyme in the pyrimidine salvage pathway [58]. Since *E. histolytica* lacks *de novo* nucleotide synthesis pathways and instead relies on the salvage pathway for nucleotides [59], EhRacM may also be involved in the regulation of nucleotide metabolism. While no current evidence directly links ribosome-associated protein quality control or pyrimidine salvage pathway with macropinocytosis regulation, protein and nucleotide synthesis is a cellular process that requires tight regulation. Thus,

investigating whether EhRacM mediates the signaling between metabolism and membrane traffic offers a fascinating area of study.

Note that we exclusively detected one DH domain-containing RhoGEF (EHI_158230) in EhRacM interactome (Table 1), which is a most likely GEF candidate of EhRacM. This Rho-GEF, which contains armadillo repeats, is most structurally similar to EhGEF2 [60]. EhGEF2 has so far been reported to be involved in erythrophagocytosis, cell proliferation, and chemo-taxis through the activation of EhRacG [61]. Further functional studies are needed to elucidate the functions of newly identified RhoGEF.

### A model of EhRacM functions in macropinocytosis

Based on the results we obtained, here we propose a hypothetical model of EhRacM in macro-pinocytosis (Fig 10). Macropinocytosis is probably initiated by the Ras-PI3K signal, which leads to actin polymerization to form a macropinocytic cup. Subsequently, the cup-enclosed

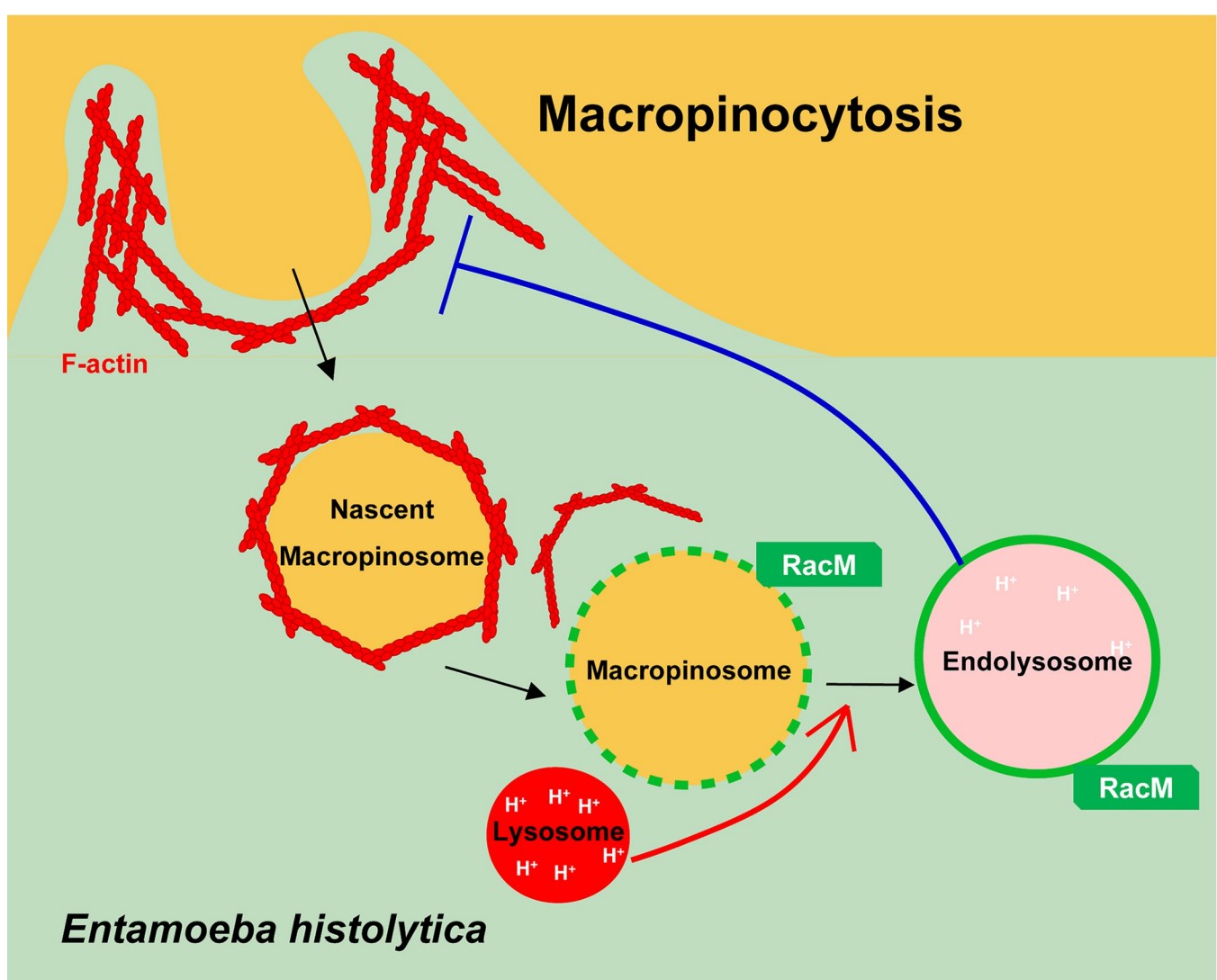

**Fig 10. Working model of the role of EhRacM in macropinocytosis.** RacM indicates EhRacM, and colored arrows indicate signals from one compartment or molecule to another. See details in the text.

nascent macropinosome is decorated with an F-actin envelope. This actin envelope is eventually removed as phosphatidylinositol 4,5-bisphosphate [PI(4,5)P2] on macropinosomes diminishes. Once uncoated, macropinosomes interact with cytoplasmic proteins, including Rab, PI3P, and V-ATPase, as well as organelles such as late endosomes and lysosomes. EhRacM is recruited to macropinosomes upon activation by RhoGEF, most likely EHI_158230, which was identified by co-IP of EhRacM. Consequently, macropinosomes (endolysosomes) get acidified, and the contents get degraded. Degradation products (nutrients) such as amino acids, nucleotides, lipids, sugars, and ions are supplied to the cytoplasm via specific transporters. We assume that nutrient signals sensed by TOR, either directly or indirectly, engage EhRacM to suppress further macropinocytosis under nutrient-sufficient conditions. The detection of Rab-related proteins in co-IP experiments suggests EhRacM's role in vesicular trafficking, which may be crucial for regulating macropinocytosis and ensuring efficient nutrient utilization. Additionally, the connection between EhRacM and the ubiquitin/proteasome pathway may link macropinosome-derived nutrients to ribosome biosynthesis.

To summarize, in this study, we identified for the first time that EhRacM is involved in macropinocytosis and cell motility in reverse directions. While EhRacM is not associated with canonical cytoskeleton regulation, it apparently regulates cytoskeletal reorganization through interactions with signaling-related proteins. Further analysis on EhRho/EhRacs and their regulatory proteins should shed light on the heterogeneous roles of Rho small GTPases in actin regulation and signaling associated with parasitism and pathogenesis of *Entamoeba*, hopefully offering unique targets for drug development.

## Material and methods

### Organisms, cultivation, and reagents

Trophozoites of *E. histolytica* clonal strains HM-1:IMSS cl6 and G3 strain were cultured axenically in 6-mL screw-capped Pyrex glass tubes in Diamond's BI-S-33 (BIS) medium at 35.5°C as previously mentioned [32,62,63]. Jurkat cells were grown at 37°C in RPMI medium (Invitrogen-Gibco, New York, USA) supplemented with 10% fetal bovine serum on a 10-cm-diameter tissue culture dish (IWAKI, Shizuoka, Japan). Phosphate buffer saline (PBS) and Rhodamine B isothiocyanate-Dextran (RITC-Dextran) were purchased from Sigma-Aldrich (Missouri, USA). Anti-GFP antibody was purchased from Merck (Mannheim, Germany). The anti-HA 16B12 monoclonal mouse antibody was purchased from Biolegend (San Diego, USA). Lipofectamine, PLUS reagent, and geneticin (G418) were purchased from Invitrogen. CellTracker™ Orange, Blue, and Deep Red were purchased from Thermo Fisher Scientific (Massachusetts, USA). Unless otherwise mentioned, restriction enzymes and DNA modifying enzymes were purchased from New England Biolabs (Massachusetts, USA). Other common reagents were from Fujifilm Wako Pure Chemical (Osaka, Japan) unless otherwise stated.

### Establishment of *E. histolytica* transformants

For antisense small RNA-mediated transcriptional silencing of ten highly expressed *Ehrho* genes [EHI_070730 (*EhracC* [18] or *Ehrho7* [25]), EHI_029020 (*Ehrho1B* [18]), XP_651936 (in the current database, EHI_013650 (*EhRacR2* [18]) and EHI_068240 (*EhracR1* [18])), EHI_129750 (*EhracG* [18] or *EhRho2* [25]), EHI_012240 (*EhracD1* [18] or *Ehrho5* [25]), EHI_197840 (*EhracA2* [18] or *Ehrho6* [25]), EHI_135450 (*EhracM* [18] or *Ehrho13* [25]), EHI_146180 (*EhracQ* [18] or *Ehrho16* [25]), EHI_194390 (*EhracJ* [18] or *Ehrho15* [25]), and EHI_192450 (*EhracP* [18] or *Ehrho14* [25])] (S1 Table), around 420-bp fragments of the protein-coding region of each gene were amplified by PCR from cDNA with sense and antisense oligonucleotides containing StuI and SacI restriction sites. The amplified product was digested

by StuI and SacI and ligated into the compatible sites of the double-digested psAP2-Gunma plasmid [23] to synthesize a gene silencing plasmid designated as psAP2-EhRacC, psAP2-EhRho1B, psAP2-XP_651936, psAP2-EhRacG, psAP2-EhRacD1, psAP2-EhRacA2, psAP2-EhRacM, psAP2-EhRacQ, psAP2-EhRacJ, and psAP2-EhRacP.

To construct a plasmid to express EhRacM with HA or GFP tag fused at the N-terminus (HA-EhRacM, HA-EhRacJ, and GFP-EhRacM), fragments of the protein-coding region of each gene were amplified by PCR from cDNA with sense and antisense oligonucleotides containing SmaI and XhoI restriction sites. The amplified product was digested with SmaI and XhoI and ligated into the compatible sites of the double-digested pEhEx-HA [64] and pEhEx-GFP vectors [65] that were predigested by SmaI and XhoI to produce pEhEx-HA-EhRacM, pEhExHA-EhRacJ, and pEhExGFP-EhRacM. Three plasmids, pEhEx-HA-EhRacM, pEhExHA-EhRacJ, and pEhExGFP-EhRacM, were introduced into the trophozoites of *E. histolytica* HM-1:IMSS cl6 strain independently, whereas psAP2-EhRacC, psAP2-EhRho1B, psAP2-XP_651936, psAP2-EhRacG, psAP2-EhRacD1, psAP2-EhRacA2, psAP2-EhRacM, psAP2-EhRacQ, psAP2-EhRacJ, and psAP2-EhRacP were introduced into G3 strain by lipofection as described previously [66]. Transformants were initially selected in the presence of 1 μg/mL G418 until the drug concentration was gradually increased to 10 μg/mL for the gene silenced stains and the HA-EhRacM, HA-EhRacJ, and GFP-EhRacM overexpressing stains. Finally, all transformants were maintained at 10 μg/mL G418 in BIS medium.

## Reverse transcriptase PCR

Reverse transcriptase PCR was performed to check mRNA levels of *EhracC*, *Ehrho1B*, *EhracG*, *EhracD1*, *EhracA2*, *EhracM*, *EhracQ*, *EhracJ*, and *EhracP* in psAP2-EhRacC, psAP2-EhRho1B, psAP2-EhRacG, psAP2-EhRacD1, psAP2-EhRacA2, psAP2-EhRacM, psAP2-EhRacQ, psAP2-EhRacJ, and psAP2-EhRacP strains respectively and mock control strain (psAP2). Total RNA was extracted from trophozoites of each strain, which was cultivated in the logarithmic phase and washed with PBS using TRIZOL reagent (Life Technologies, California, USA). Approximately 5 μg of DNase-treated total RNA was used for cDNA synthesis using Superscript III First-Strand Synthesis System (Thermo Fisher Scientific) with reverse transcriptase and oligo (dT) primer according to the manufacturer's protocol. ExTaq PCR system was used to amplify DNA from the cDNA template using the primer pairs listed in S2 Table. The PCR conditions were as follows: initial denaturation at 94°C for 1 minute; then 35 cycles at 98°C for 10 sec, 55°C for 15 sec, and 72°C for 15 sec; and a final extension at 72°C for 1.5 minutes. The PCR products obtained were resolved by agarose gel electrophoresis.

## Quantitative real-time (qRT) PCR

The relative mRNA levels of *EhracC*, *Ehrho1B*, *EhracG*, *EhracD1*, *EhracA2*, *EhracM*, *EhracQ*, *EhracJ*, *EhracP*, and *RNA polymerase II* gene (EHI_056690), as an internal standard, were measured by qRT-PCR. The PCR reaction was prepared using Fast SYBR Master Mix (Applied Biosystems, California, USA) with cDNA and a primer set shown in the S2 Table. PCR was conducted using the StepOne Plus Real-Time PCR system (Applied Biosystems) with the following cycling conditions: An initial step of denaturation at 95°C for 20 sec, followed by 40 cycles of denaturation at 95°C for 3 sec, annealing and extension at 60°C for 30 sec. The mRNA expression level of each *Ehrho* gene in the transformants was presented as relative to that in the control transfected with psAP2.

## Macropinocytosis efficiency evaluation by FACS (gene silenced strains)

Trophozoites of *EhracM* and *EhracJ* gene silenced, and mock control (psAP2) strains were incubated in BIS medium containing 2 mg/mL of RITC-dextran for 15, 30, 60, 90, and 120 minutes and washed three times with pre-cold PBS. Measurement was performed by BD Accuri™ C6 Plus Flow Cytometer (Becton Dickinson, New Jersey, USA).

For the FACS analysis, first, gates were designed to distinguish debris from amoeba cells by 2-dimensional scatter plots of the forward scatter (FSC, which reflects cell sizes) and the side scatter (SSC, which reflects internal cellular complexities). FlowJo software (Becton Dickinson) was used for the data analysis. Afterward, the geometric mean value of fluorescence intensity in the PE-A channel was analyzed. The change of the geometric mean value at each time point was compared to the original value and normalized by the value of psAP2 mock control strain at 120 minutes. Statistical significance was examined with an unpaired t-test based on three independent trials in each assay.

## Phagocytosis and trogocytosis efficiency evaluation by FACS (gene silenced strains)

Trophozoites of *EhracM* gene silenced and mock control (psAP2) strains were co-incubated with 2 μm carboxylated beads (phagocytosis), heat-killed Jurkat cells (phagocytosis), 10 μm carboxylated beads (phagocytosis), or live Jurkat cells (trogocytosis) at the ratio of 1 to 60, 1 to 5, 1 to 5, and 1 to 5, respectively. The maximum incubation times were also 60, 15, 60, and 60 minutes, respectively. Amoeba cells were washed with pre-cold 2% galactose-PBS after the co-incubation. Measurement was performed by BD Accuri™ C6 Plus Flow Cytometer.

For the FACS analysis, first, gates were designed to distinguish debris, Jurkat cells, or beads from amoeba cells by 2-dimensional scatter plots of the FSC and SSC. FlowJo software was used for the data analysis. For bead and heat-killed Jurkat phagocytosis measurement, compared to the original fluorescence intensity of amoeba cells, the percentage of amoeba cells containing bead(s) or Jurkat cell(s) was estimated based on the increase of fluorescence intensity in the PE-A channel. These percentages were normalized by the value of psAP2 mock control strain at 60 or 15 minutes (60 minutes for 2 μm or 10 μm carboxylated bead, 15 minutes for heat-killed Jurkat cell). On the other hand, as for live Jurkat trogocytosis measurement, the geometric mean value of fluorescence intensity in the PE-A channel was analyzed. The change of the geometric mean value at each time point was compared to the original value and then normalized by that of psAP2 mock control strain at 60 minutes. Statistical significance was examined with an unpaired t-test based on three independent trials in each assay.

## Macropinocytosis efficiency evaluation by FACS (GFP-EhRacM overexpressing strains)

Trophozoites of the GFP-EhRacM overexpressing and mock control (pEhEx-GFP) strains were incubated in BIS medium containing 2 mg/mL of RITC-dextran for 15, 30, 60, 90, and 120 minutes. Trophozoites were then washed three times with pre-cold PBS. Measurements were performed by BD Accuri™ C6 Plus Flow Cytometer.

For the FACS analysis, first, gates were designed to distinguish debris from amoeba cells by 2-dimensional scatter plots of FSC and SSC as in the gene silencing assay. FlowJo software was used for the data analysis. After screening cells that overexpress GFP-EhRacM with fluorescence intensity in FITC-A channel, compensation between FITC-A and PerCP-A channels was conducted by the AutoSpill compensation function of the FlowJo software. Next, the PerCP-A channel fluorescence intensity of the population in the set gate was analyzed. In this

study, dextran's fluorescence intensity was stronger than amoeba cells in the PerCP-A channel. We compared the change of the geometric mean value at each time point to the original value and then normalized them by that of pEhEx-GFP mock control strain at 120 minutes. Statistical significance was examined with an unpaired t-test based on three independent trials.

## Dextran release (exocytosis) evaluation by FACS

Trophozoites of *EhracM* gene silenced and mock control (psAP2) strains were incubated in BIS medium containing 2 mg/mL of RITC-dextran for 15 minutes. Afterward, the medium was replaced with pre-warmed dextran-free BIS medium and chased for 0, 10, 30, and 60 minutes. Trophozoites were then washed three times with pre-cold PBS. Measurement was performed by BD Accuri™ C6 Plus Flow Cytometer.

For the FACS analysis, first, gates were designed to distinguish debris from amoeba cells by 2-dimensional scatter plots of FSC and SSC. FlowJo software was used for the data analysis. The fluorescence intensity relative to the original intensity (time point 0, 100%) in each trial was averaged. Statistical significance was examined with an unpaired t-test based on three independent trials in each assay.

## Cell diameter measurement

For the cell diameter measurement, around $1 \times 10^3$ trophozoites of *EhracM* gene silenced, *EhracJ* gene silenced, and psAP2 mock control strains were cultivated in the logarithmic phase and washed with PBS. Next, trophozoites were resuspended with 100 μL of PBS, and 5 μL was taken and diluted with 10 mL of CASY ton buffer (Omni Life Science, Bremen, Germany). The mean cell diameter of each strain was calculated based on over 1000 viable trophozoites using CASY (Omni Life Science). The mean cell volume of each strain was estimated based on the mean cell diameter.

## Motility assay

Trophozoites of *EhracM* gene silenced and the psAP2 mock control strains were incubated in Opti-MEM (Thermo Fisher Scientific) containing 10 μM of CellTracker™ Deep Red Dye for 40 minutes at 35.5˚C. The trophozoites were then washed twice with PBS and resuspended with BIS medium. Next, the trophozoites were transferred into 6 ml screw-capped Pyrex glass tubes and incubated for 30 minutes at 35.5˚C. The BIS medium was then swapped with another filtered BIS medium, and the detached trophozoites were removed. The trophozoites were then placed on ice for 5 minutes and centrifuged at $300 \times G$ for 5 minutes. After the removal of the supernatant, pellets were resuspended with filtered BIS medium, which does not contain any debris to observe with a clear field of view, and incubated in EZVIEW™ Glass Bottom Assay Plates (96 well) (IWAKI) at 35.5˚C for 30 minutes. Plates were then transferred into a CQ1 confocal quantitative image cytometer (Yokogawa Electric Co., Ltd., Tokyo, Japan), and images were taken under the following conditions: 37˚C, one second of an interval, 100 times of duration. The obtained data were analyzed by CellPathfinder image analysis software (Yokogawa Electric Co., Ltd.). Each value was calculated based on the following formula.

$$Total\ Distance = \sum_{i=1}^{99} (p_{i+1} - p_i)$$

$$Net\ Distance = p_{100} - p_1$$

$$Maximum\ Distance = \max(p_{i+1} - p_i)$$

$$Confinement\ Ratio = \frac{Net\ Distance}{Total\ Distance}$$

$$Average\ Speed = \frac{1}{99}\sum_{i=1}^{99} v_i \left( v_i = \frac{|p_{i+1} - p_i|}{\Delta t} \right)$$

$$Mean\ Net\ Distance\ Speed = \frac{Net\ Distance}{100}$$

$$Linearity\ of\ Forward\ Progression = \frac{Mean\ Net\ Distance\ Speed}{Average\ Speed}$$

$i$: frame number, $p$: displacement coordinate, $\Delta t$: time interval, $v_i$: autocorrelated velocity vector between two subsequent time frames.

## Immunoblot analysis

Trophozoites of amoeba transformants expressing HA-EhRacM, HA-EhRacJ, or GFP-Eh-RacM growing in the exponential growth phase were collected and washed with PBS. After resuspension in lysis buffer (50 mM Tris-HCl, pH 7.5, 150 mM NaCl, 1% Triton-X 100 (ICN Biochemicals, Inc, Ohio, USA), 0.5 mg/mL E-64 (Peptide Institute, Inc., Osaka, Japan), and cOmplete™ Mini, EDTA-free Protease Inhibitor (Merck, Mannheim, Germany)), the trophozoites were kept on ice for 10 minutes, followed by centrifugation at $16,000 \times G$ for 5 minutes. Approximately 30 µg of the total cell lysates were separated on 15% (HA-EhRacM and HA-Eh-RacJ) or 10% (GFP-EhRacM) SDS-PAGE and subsequently electrotransferred onto nitrocellulose membranes. The membranes were incubated in 10% skim milk in Tris-Buffered Saline and Tween-20 [TBS-T; 50 mM Tris-HCl, pH 8.0, 150 mM NaCl, and 0.05% Tween-20 (TCI, Tokyo, Japan)] for 30 minutes at room temperature to block non-specific protein. The blots were reacted with one of the following primary antibodies diluted as indicated: anti-HA 16B12 monoclonal mouse antibody (1:1,000), anti-GFP mouse monoclonal antibody (1:100), and anti-CS1 rabbit polyclonal antisera [67] (1:1,000) at 4°C overnight. The membranes were washed with TBS-T and further reacted with horseradish peroxidase-conjugated (HRP) anti-mouse (Invitrogen) (1:1,000) or anti-rabbit IgG antisera (Thermo Fisher Scientific) (1:10,000) at room temperature for 1 hr. After washings with TBS-T, the specific proteins were visualized with a chemiluminescence HRP Substrate system (Merck) using ChemiDoc Imaging System (Biorad, California, USA) according to the manufacturer's protocol.

## Indirect immunofluorescence assay (IFA)

Trophozoites of the transformant strain expressing HA-EhRacM and HA-EhRacJ were suspended with BIS medium and transferred to a Millicell EZ SLIDE 8 well glass slide (Merck) to culture. After one hour of incubation in an anaerobic chamber set to 35.5°C, BIS medium was gently removed, and cells were fixed with PBS containing 3.7% paraformaldehyde (Kanto Chemical Co., Inc., Tokyo, Japan) at room temperature for 10 minutes and subsequently gently washed twice by PBS. The cells were then permeabilized with 0.2% Saponin (Sigma-Aldrich) in 1% bovine serum albumin (BSA) (Sigma-Aldrich) for 10 minutes at room temperature. Next, the cells were incubated with anti-HA mouse monoclonal antibody (1:1,000) for 1

hour at room temperature. After washing twice with 0.1% BSA in PBS solution (BSA-PBS), the sample reacted with Alexa Fluor-488 conjugated anti-mouse IgG (1:1,000) (Thermo Fisher Scientific) for 1.5 hours at room temperature. The sample was washed twice with 0.1% BSA-PBS and covered with mounting medium [1 mg/mL p-phenylenediamine (Sigma-Aldrich) in a mixture of 90% glycerol and PBS] and 24 mm×60 mm coverslip (Matsunami Glass IND., LTD, Osaka, Japan). The images were captured by Andor Dragonfly 200 Spinning Disk Confocal Microscope System (Andor, Northern Ireland, UK) using a 60 × oil immersion objective with default settings and analyzed by IMARIS software (Oxford Instruments, Abingdon-on-Thames, UK). The fluorescence intensity was measured by the "Plot Profile" function from Fiji-ImageJ software [68].

## Live cell imaging and macropinocytosis live imaging

Trophozoites of the transformant strain expressing GFP-EhRacM were suspended with 1 mL of filtered BIS medium, which does not contain any debris, and cultured on a glass-bottom dish (MatTek Corporation, Massachusetts, USA) at 35.5˚C for 30 minutes. The central part of the dish was then carefully covered with an 18-mm square coverslip (Muto Pure Chemicals Co., LTD, Japan, Tokyo). For the macropinocytosis live imaging, before covering with a coverslip, BIS medium was gently removed and replaced with BIS medium containing RITC-dextran. Live images were captured by Andor Dragonfly 200 Spinning Disk Confocal Microscope System using a 60 × oil immersion objective with default settings on the time series mode and analyzed by IMARIS software. The fluorescence intensity was measured by the "Plot Profile" function from Fiji-ImageJ software [68].

## Phalloidin staining

Trophozoites of the transformant strain expressing GFP-EhRacM were suspended with BIS medium and transferred to a Millicell EZ SLIDE 8-well glass slide to culture. After one hour of incubation in an anaerobic chamber at 35.5˚C, BIS medium was gently removed, and cells were incubated with fixation solution containing 10 mM PIPES (pH 7.4), 3 mM MgCl$_2$, 1 mM EGTA (pH 8), 1 mM DTT, and 4% paraformaldehyde at 35.5˚C for 30 minutes and subsequently permeabilized with 0.05% Triton for one minute at room temperature. The cells were then washed with PBS and quenched autofluorescence and reactive aldehydes after fixation with PBS containing 50 mM of NH$_4$Cl for 15 minutes at room temperature. The cells were subsequently blocked with 2% BSA-PBS for 1 hour at room temperature and then stained with 5 μU/mL of phalloidin 568 (Sigma-Aldrich) for 1 hour at room temperature. After washing with PBS, cells were covered with mounting medium [1 mg/mL p-phenylenediamine in a mixture of 70% glycerol and PBS]. The images were captured by Andor Dragonfly 200 Spinning Disk Confocal Microscope System using a 60 × oil immersion objective with default settings and analyzed by IMARIS software.

## Double-staining of SiR-Actin and phalloidin

Trophozoites of *E. histolytica* HM-1: IMSS cl6 strain were suspended with BIS medium and transferred to a Millicell EZ SLIDE 8-well glass slide to culture. After the incubation in an anaerobic chamber at 35.5˚C, BIS medium was gently removed, and cells were fixed with fixation solution containing 10 mM PIPES (pH 7.4), 3mM MgCl$_2$, 1 mM EGTA (pH 8), 1 mM DTT, and paraformaldehyde at 35.5˚C for 30 minutes and subsequently permeabilized with 0.05% Triton for one minute at room temperature. The cells were then washed with PBS and quenched autofluorescence and reactive aldehydes after fixation with PBS containing 50 mM of NH$_4$Cl for 15 minutes at room temperature. The cells were subsequently blocked with 2%

BSA-PBS for 1 hour at room temperature and then stained with PBS containing 10 U/mL phalloidin 488 (Sigma-Aldrich), 10 M/mL SiR-Actin (Cytoskeleton, Inc., Colorado, USA) and 10 M/mL Verapamil (Cytoskeleton, Inc.) for 1 hour at room temperature. After washing with PBS, cells were covered with mounting medium [1 mg/mL p-phenylenediamine in a mixture of 70% glycerol and PBS]. The images were captured by confocal microscope Zeiss LSM 780 (Zeiss, Oberkochen, Germany) with a 63 × oil immersion objective with default settings and analyzed by ZEN software (Zeiss).

## Macropinocytosis live imaging stained by SiR-Actin

Trophozoites of the transformant strain expressing GFP-EhRacM were suspended with 1 mL of filtered BIS medium containing 10 M/mL SiR-Actin and 10 M/mL Verapamil and cultured on a glass-bottom dish for 1 hour at 37˚C. RITC-dextran containing BIS medium was added to the central part of the dish and then carefully covered with an 18 mm square coverslip. Live images were captured by Andor Dragonfly 200 Spinning Disk Confocal Microscope System using a 60 × oil immersion objective with default settings on the time series mode and analyzed by IMARIS software. The fluorescence intensity was measured by the "Plot Profile" function from Fiji-ImageJ software [68].

## Subcellular fractionation

Approximately 200 mg of amoeba cells were washed with cold PBS containing 2% D(+)-galactose, resuspended in homogenization buffer (250 mM sucrose, 50 mM Tris HCl pH 7.5, 50 mM NaCl, 0.1 mg/mL of E64, and cOmplete™ Mini EDTA-free Protease Inhibitor) and homogenized on ice with 120 strokes by a Dounce homogenizer with a tight-fitting pestle. After unbroken cells were removed by centrifugation at 400 × G for 2 minutes, the supernatant (Ho) was centrifuged at 13,000 × G at 4˚C for 10 minutes to obtain the pellet (P13) and supernatant fractions. The supernatant fraction was washed with homogenization buffer. The S13 fraction was further separated by centrifugation at 100,000 × G at 4˚C for 1 hour to obtain soluble (S100) fractions and pellet (P100). The P100 fraction was subsequently washed with homogenization buffer at 100,000 × G at 4˚C for 1 hour. These fractions were subjected to immunoblot analyses with anti-HA, anti-GFP, anti-CS1 [67], or anti-heavy subunit of cell surface Gal/GalNAc lectin (HgL) (3F4) (37) antibodies.

## Cross-linking and co-immunoprecipitation of HA-tagged EhRacM and EhRacJ

Trophozoites of the amoeba transformants overexpressing HA-EhRacM, HA-EhRacJ, pEhEx-HA (mock control) were cultured in a 10-cm-diameter tissue culture dish (IWAKI) with BIS medium under an anaerobic condition using Anaerocult® A (Merck). Trophozoites were detached from the surface of the dishes by adding cold PBS into the dishes and incubating them at 4˚C for 10 minutes. After washing with PBS three times, the cell pellets were resuspended in 500 μL of 8 mg/mL dithiobis (succinimidyl propionate; DSP) solution (Thermo Fisher Scientific) and cross-linked. The mixture was incubated on the rotator (10 rpm) at RT for 30 minutes. To quench the reaction, 50 μL of 1 M Tris-HCl (pH 7.5) was added, and the mixture was further incubated as above for 15 minutes at RT. After the pellets were treated with DSP, they were washed twice with PBS. The cells were lysed with 1 mL of lysis buffer [50 mM Tris-HCl (pH 7.5), 150 mM NaCl, containing 1% Triton X-100, 0.05 mg/mL E-64, and cOmplete™, Mini, EDTA-free Protease Inhibitor)]. After the debris was removed by centrifugation at 16,000 × G at 4˚C for 10 minutes, the lysates were mixed and incubated with 50 μL of Protein G Sepharose 4 Fast Flow (Cytiva, Uppsala, Sweden) at 4˚C for 1 hour. After

centrifugation at 800 × G for 3 minutes at 4˚C, the supernatant was mixed with 50 μL of Monoclonal Anti-HA-Agarose antibody produced in mouse, clone HA-7, purified immunoglobulin conjugated to agarose beads (Sigma-Aldrich) and incubated with inversion at 4˚C for 3.5 hours. The mixture was then centrifuged at 800 × G for 3 minutes at 4˚C to separate unbound proteins. Afterward, the beads were washed three times with 1 mL of lysis buffer. The obtained beads fraction was resuspended with 180 μL of lysis buffer containing Influenza Hemagglutinin (HA) -peptide (Sigma-Aldrich) at the final concentration of 0.2 mg/mL. The mixture was incubated at 4˚C overnight to elute the bound proteins for HA-EhRacM or HA-EhRacJ. After centrifugation at 800 × G for 3 minutes at 4˚C, the supernatant was collected and submitted to Mass Spectrometry and Proteomics Core Facility, Johns Hopkins University School of Medicine, Baltimore, MD, USA, for mass spectrometry analysis.

## Mass spectrometry

Protein extracts were buffer exchanged using SP3 paramagnetic beads (GE Healthcare) [69]. Samples were briefly rehydrated in 100 μL 10 mM Triethylammonium bicarbonate (TEAB) + 1% SDS, and disulfide bonds were reduced with 10 μL of 50 mM dithiothreitol for 1 hour at 60˚C. Samples were cooled to RT and pH adjusted to ~7.5, followed by alkylation with 10 μL of 100 mM iodoacetamide in the dark at RT for 15 minutes. Next, 100 μg (2 μL of 50 μg/μL) SP3 beads were added to the samples, followed by 120 μL 100% ethanol. Samples were incubated at RT and shaken for 5 minutes. Following protein binding, beads were washed three times with 180 μL 80% ethanol. Proteins were digested on-bead with trypsin (Pierce) at 37˚C overnight (1 μg enzyme). The supernatant was removed from the beads and dried, followed by rehydration in 2% acetonitrile/0.1% formic acid prior to mass spectrometry analysis.

Peptides were analyzed by reverse-phase chromatography-tandem mass spectrometry on an EasyLC1100 UPLC interfaced with an Orbitrap Fusion Lumos mass spectrometer (Thermo Fisher Scientific). Peptides were separated using a 0%–100% acetonitrile in 0.1% formic acid gradient over 90 minutes at 300 nL/minute. The 75 μm ×15 cm column (PicoFrit Self-pack emitter, New Objective) was packed in-house with ReproSIL-Pur-120-C18-AQ (3 μm, 120 Å bulk phase, Dr. Maisch). Survey scans of precursor ions were acquired from 350–1800 m/z at 120,000 resolution at 200 m/z, automatic gain control (AGC) of $4 \times 10^5$, and an RF lens setting of 45% and internal mass calibration. Precursor ions were individually isolated within 1.5 m/z by data-dependent acquisition with a 15-second dynamic exclusion and fragmented using an HCD activation with a collision energy of 30. Fragment ions were analyzed at 30,000 resolution and AGC of $1 \times 10^5$.

All data files were analyzed using Mascot (Matrix Science, London, UK; version 2.8.2) with the UniProt_E_histolytica_txid294381 database (20546 entries), assuming that the digestion enzyme was trypsin. Mascot was searched with a fragment ion mass tolerance of 10.0 PPM and a parent ion tolerance of 5.0 PPM. Carbamidomethylation of cysteine was specified as a fixed modification. Deamidation of asparagine and glutamine, oxidation of methionine, and formylation of lysine and the protein N-terminus were listed as variable modifications.

Peptide identifications from the Mascot searches were processed and imported into Scaffold (Proteome Software Inc.), with peptide validation (1% false-discovery rate) and protein inference (95% confidence) by PeptideProphet [70] and ProteinProphet [71], respectively. The mass spectrometry proteomics data have been deposited to the ProteomeXchange Consortium via the PRIDE [72] partner repository with the dataset identifier PXD042282 and PXD051913 (For EhRacM and EhRacJ, respectively).

The Gene Ontology (GO) analysis was conducted by PANTHER Overrepresentation Test (Released 20230705) with Fisher's Exact test, based on the GO Ontology database DOI: 10.5281/zenodo.7942786. The plots were mapped by the ggplot2 R package (3.5.1).

### Growth assay of *E. histolytica* trophozoites

Approximately 50,000 trophozoites of *EhracM* gene silenced and psAP2 mock control strains grown in the logarithmic phase were incubated in 6 ml screw-capped Pyrex glass tubes BIS medium containing 10 μg/mL G418 at 35.5°C. The parasites were counted every 24 hours. The concentration of trophozoites estimated the doubling time after 24 and 48 hours.

### RNA-seq

Total RNA was extracted from trophozoites of each strain (*EhracM* gene silenced, *EhracJ* gene silenced, and psAP2 mock control strains) by using TRIZOL reagent according to the manufacturer's instructions. The obtained total RNA was subjected to library construction using a directional library preparation protocol and sequenced on an Illumina Novaseq 6000 (Novogene, China). Sequence reads were trimmed, mapped, and assembled to the reference genome assembly (JCVI-ESG2-1.0). Then, FeatureCounts v1.5.0-p3 was used to count the read numbers mapped to each gene. Then, fragments per kilobase of exon per million mapped reads (FPKM) of each gene were calculated based on the length of the gene and the read count mapped to this gene. Differential expression analysis of two conditions/groups (three biological replicates per condition) was performed using the DESeq2 R package (1.20.0). The resulting P-values were adjusted using Benjamini and Hochberg's approach to control the false discovery rate. Genes with an adjusted P-value $<=0.05$ found by DESeq2 were assigned as differentially expressed. Volcano plots were mapped by the ggplot2 R package (3.5.1) and ggrepel R package (0.9.5). Gene Ontology (GO) enrichment analysis was performed using the clusterProfiler R package (3.8.1) and mapped by the ggplot2 R package. The data discussed in this publication have been deposited in NCBI's Gene Expression Omnibus [73] and are accessible through GEO Series accession number GSE229171 (https://www.ncbi.nlm.nih.gov/geo/query/acc.cgi?acc=GSE229171).

### Graphs and statistical analyses

Line plots and bar graphs in this paper were generated in GraphPad Prism 9.4.1. All the analyses were done in GraphPad Prism version 9.4.1. Detailed statistical methods are described in each figure.

### Multiple sequence alignment of Rho small GTPases

Multiple sequence alignment was conducted by Clustal Omega [74]. The output was visualized by Jalview software (2.11.3.3) [75].

### Supporting information

**S1 Fig. Alignment of representative Rho small GTPases in *Homo sapiens* and *Entamoeba histolytica*.** The amino acid sequences of HsRhoA (NP_001300870.1), HsRac1 (NP_008839.2), HsCdc42 (NP_001034891.1), EhRho1B (EHI_029020), EhRacA2 (EHI_197840), EhRacD1 (EHI_012240), EhRacG (EHI_129750), and EhRacM (EHI_135450) were aligned. The conserved G boxes are highlighted in red, while the Rho insert regions are highlighted in blue. (TIF)

**S2 Fig. Gene silencing of *EhracM* did not affect fluid-phase marker release efficiency.** Trophozoites of psAP2 mock and *EhracM* gene silenced (gs) strains were incubated in BIS medium containing RITC dextran containing for 15 minutes. Amoeba cells were then incubated in dextran-free BIS medium and chased for 0, 10, 30, and 60 minutes. The fluorescence intensity of each amoeba cell was measured by FACS as described in Materials and Methods. The fluorescence intensity relative to the original intensity (time point 0 min, 100%) in each trial was averaged. Statistical significance was examined with an unpaired t-test (ns: not significant). Error bars indicate standard deviations of three biological replicates.
(TIF)

**S3 Fig. EhRacM is a negative regulator of small beads phagocytosis.** (A), (D) Trophozoites of psAP2 mock and *EhracM* gene silenced (gs) strains were incubated with 2-μm (A) or 10-μm (D) carboxylated polystyrene beads. The fluorescence intensity of each amoeba cell was measured by FACS as described in Materials and Methods. The relative percentages of amoeba cells containing bead(s) are shown. Each value is standardized by the value of psAP2 mock control strain at 60 min. (B) Trophozoites of psAP2 mock and *EhracM* gs strains were incubated with pre-killed Jurkat cells, which were heated at 55˚C for 15 min. The fluorescence intensity of each amoeba cell was measured by FACS as described in Materials and Methods. The relative percentages of amoeba cells containing Jurkat cell(s) are shown. Each value is standardized by the value of psAP2 mock control strain at 15 min. (C) Trophozoites of psAP2 mock and *EhracM* gs strains were incubated with live Jurkat cells. The fluorescence intensity of each amoeba cell was measured by FACS as described in Materials and Methods. The relative increase of geometric mean in PE-A channel is shown. Each value is standardized by the value of psAP2 mock control strain at 60 min. Statistical significance was examined with unpaired t-test (*p<0.05, ns: not significant). Error bars indicate standard deviations of three biological replicates.
(TIF)

**S4 Fig. Gene silencing of *EhracM* did not affect cell proliferation.** Doubling time of the psAP2 mock and *Ehrac* gene silenced (gs) strains. Statistical significance was examined with unpaired t-test (ns: not significant). Error bars indicate standard deviations of three biological replicates.
(TIF)

**S5 Fig. GO enrichment analysis of differentially expressed genes in *EhracM* gene silenced strain (molecular function (MF)).** The result of GO enrichment analysis for downregulated (A) or upregulated (B) genes in *EhracM* gene silenced strain. All the differential genes with adjusted p-value (padj) < 0.05 were included in this analysis regardless of their foldchange compared to those of mock strain. GO terms that showed padj < 0.05 were selected in descending order of Gene Ratio (Ratio between the number of differentially expressed genes in each GO term and all differentially expressed genes that can be found in GO database) for each entry. Each dot size reflects the count size, whereas its color reflects the padj. The x-axis indicates Gene Ratio.
(TIF)

**S6 Fig. Differentially expressed genes in *EhracM* gene silenced strain.** Volcano plots of the total RNA expression level of *EhracM* gene silenced strain vs psAP2 mock strain based on three biological replicates are shown. The x-axis reflects the folding change of total RNA amount in *EhracM* gene silenced strain compared with the psAP2 mock strain, whereas the y-axis shows the adjusted p-value (padj). Upregulated genes are shown in red, whereas

downregulated genes are shown in blue. Others are shown in gray.
(TIF)

**S7 Fig. Differentially expressed genes in *EhracJ* gene silenced strain.** Volcano plots of the total RNA expression level of *EhracJ* gene silenced strain vs psAP2 mock strain based on three biological replicates are shown. The x-axis reflects the folding change of total RNA amount in *EhracJ* gene silenced strain compared with psAP2 mock strain, whereas the y-axis shows the adjusted p-value (padj). Upregulated genes are shown in red, whereas downregulated genes are shown in blue. Others are shown in gray.
(TIF)

**S8 Fig. Cellular localization of EhRacJ in *E. histolytica* trophozoite.** (A) Immunoblot detection of HA-EhRacJ in *E. histolytica* transformants. Approximately 30 µg of total lysates from mock-transfected control (pEhEx-HA) and HA-EhRacJ expressing transformants were subjected to SDS-PAGE and immunoblot analysis using anti-HA and anti-CS1 (loading control) polyclonal antibodies. The arrows indicate the approximate sizes of HA-EhRacJ. (B) The immunofluorescence image of HA-EhRacJ expressing trophozoite (left) and the line intensity plot along with the yellow arrow (right).
(TIF)

**S9 Fig. Representative image of trophozoite expressing GFP-EhRacM.** Representative image of live imaging of GFP-EhRacM (green) expressing trophozoite. GFP-EhRacM localizes mostly at the cytosol and sometimes plasma membrane (arrow) but enriches at the small vesicle surface (arrowheads). Cell Tracker Blue staining indicates the whole cytosol (blue).
(TIF)

**S10 Fig. Overexpression of EhRacM decreased dextran macropinocytosis efficiency.** Trophozoites of GFP (mock) and GFP-EhRacM overexpressing strains were incubated in RITC dextran containing BIS medium to evaluate macropinocytosis. The fluorescence intensity of amoeba cells was measured by FACS as described in Material and Methods. Each value is standardized by the value of pEhEx-GFP mock control strain at 120 min. Statistical significance was examined with unpaired t-test (ns: not significant). Error bars indicate standard deviations of three biological replicates.
(TIF)

**S11 Fig. Validation of F-actin staining by SiR-Actin.** Trophozoites of *E. histolytica* HM-1: IMSS cl6 strain were double stained by phalloidin (green) and SiR-Actin (magenta). Each row shows representative actin-rich structures.
(TIF)

**S12 Fig. GFP-EhRacM was gradually recruited to macropinosomes after the removal of the actin envelope.** Montage of live imaging time series of a representative GFP-EhRacM expressing trophozoite in which macropinocytosis was monitored. F-actin was visualized using SiR-Actin (magenta), while GFP-EhRacM expressing trophozoites were shown in green (second row). The third-row panels indicate inverted grayscale signals of GFP-EhRacM. The trophozoites were incubated with RITC dextran (red). The white arrowheads indicate the site of macropinosome formation and the resultant macropinosome. The fourth-row panels show the GFP-EhRacM's fluorescence intensity plot along with the trajectory of yellow arrows depicted in the corresponding third-row panels. Macropinosome areas are highlighted in red, and the mean GFP-EhRacM's signal intensity at the initial macropinocytic cup (approximately 27) is shown in green. Bars, 5 µm. The F-actin envelope dissociation was captured at 50.662 [s].
(TIF)

**S13 Fig. Identified hits from HA-EhRacM co-IP.** (A) Venn diagram of hit proteins in three independent co-IPs. Hits were defined as proteins whose QV was higher in HA-EhRacM than mock. Hits from the first co-IP are shown in red, hits from the second in blue, and hits from the third in green. 107 hits were identified in common in the three co-IPs, which are listed in S9 Table. (B) STRING protein-protein interaction network for the 107 proteins identified in common across the three co-IPs. Network nodes represent the proteins. Known interactions from curated databases are shown in sky blue, from experimentally determined are shown in magenta. Predicted interactions based on gene neighborhood are shown in green, based on gene fusions are in red, and gene co-occurrence is in blue. Moreover, textmining interactions are shown in yellow, co-expression proteins are tied in black, and protein homologies are shown in light blue. Proteins in the proteasome pathway (count in the network: 13 of 39, false discovery rate: 6.64e-12), the ribosome pathway (count in the network: 10 of 145, false discovery rate: 0.0016), and the pyrimidine metabolism (count in the network: 4 of 23, false discovery rate: 0.0116) in KEGG are shown as red, blue, and green nodes, respectively.
(TIF)

**S14 Fig. GO enrichment analysis of EhRacJ binding protein candidates identified by HA-EhRacJ co-IP.** The results of PANTHER GO enrichment analysis on hit proteins from HA-EhRacJ co-IP. Proteins were classified by biological process (BP) (A) and molecular function (MF) (B). GO terms whose FDR-corrected p-value (FDR) are smaller than 0.05 are shown in descending order of FDR value for each entry (at most 20 terms). Each dot size reflects the count size, whereas its color reflects the FDR. The x-axis indicates fold change.
(TIF)

**S15 Fig. Structual prediction of EhRacM and EhRacJ by Alphafold2.** The predicted structures of EhRacM (C4M9S3) (A) and EhRacJ (C4LZV2) (B) were obtained from the AlphaFold Protein Structure Database (https://alphafold.ebi.ac.uk). N indicates the N-terminus, whereas C indicates the C-terminus of each protein. Each color represents the per-residue model confidence score (pLDDT), ranging from 0 to 100. Blue indicates a pLDDT greater than 90, light blue signifies scores between 70 and 90, yellow denotes scores between 50 and 70, and orange represents scores less than 50.
(TIF)

**S1 Table. 10 *Ehrho* genes targeted for silencing.** List of amino acid sequences and annotation names of 10 *Ehrho* (*Ehrac*) genes targeted for silencing in this study.
(XLSX)

**S2 Table. List of primers used in this study.** Restriction sites are marked by bold letters. F stands for a forward primer, whereas R stands for a reverse primer.
(XLSX)

**S3 Table. List of the genes downregulated in *EhracM* gene silenced strain.** List of the genes whose total RNA expression level decreased more than two times in the *EhracM* gene silenced strain (*EhracM* gs) compared to the psAP2 mock strain. Only the genes with adjusted p-value (padj) < 0.05 are shown. The 6 rows on the left indicate the RNA expression level of each sample. 1st, 2nd, and 3rd indicate three biologically independent replicates. The *EhracM* gs Average column indicates the transcript amount of each gene in the *EhracM* gs strain, whereas the psAP2 Average column shows that of the psAP2 mock strain, respectively, based on three biological replicates. Each gene was classified by Protein ANalysis THrough Evolutionary Relationships (PANTHER) classification system's protein class annotation data set. Rho-related or

Ras-related genes are shown in orange.
(XLSX)

**S4 Table. List of the genes upregulated in *EhracM* gene silenced strain.** List of the genes whose total RNA expression level increased more than two times in the *EhracM* gene silenced strain (*EhracM* gs) compared to the psAP2 mock strain. Only the genes with adjusted p-value (padj) < 0.05 are shown. The 6 rows on the left indicate the RNA expression level of each sample. 1st, 2nd, and 3rd indicate three biologically independent replicates. The *EhracM* gs Average column indicates the transcript amount of each gene in *EhracM* gs strain, whereas the psAP2 Average column shows that of the psAP2 mock strain, respectively, based on three biological replicates. Each gene was classified by PANTHER classification system's protein class annotation data set. Rho-related or Ras-related genes are shown in orange.
(XLSX)

**S5 Table. List of the genes downregulated in *EhracJ* gene silenced strain.** List of the genes whose total RNA expression level decreased more than two times in the *EhracJ* gene silenced strain (*EhracJ* gs) compared to the psAP2 mock strain. Only the genes with adjusted p-value (padj) < 0.05 are shown. The 6 rows on the left indicate the RNA expression level of each sample. 1st, 2nd, and 3rd indicate three biologically independent replicates. The *EhracJ* gs Average column indicates the transcript amount of each gene in the *EhracJ* gs strain, whereas the psAP2 Average column shows that of the psAP2 mock strain, respectively, based on three biological replicates. Each gene was classified by PANTHER classification system's protein class annotation data set.
(XLSX)

**S6 Table. List of the genes upregulated in *EhracJ* gene silenced strain.** List of the genes whose total RNA expression level increased more than two times in the *EhracJ* gene silenced strain (*EhracJ* gs) compared to the psAP2 mock strain. Only the genes with adjusted p-value (padj) < 0.05 are shown. The 6 rows on the left indicate the RNA expression level of each sample. 1st, 2nd, and 3rd indicate three biologically independent replicates. The *EhracJ* gs Average column indicates the transcript amount of each gene in the *EhracJ* gs strain, whereas the psAP2 Average column shows that of the psAP2 mock strain, respectively, based on three biological replicates. Each gene was classified by PANTHER classification system's protein class annotation data set.
(XLSX)

**S7 Table. Expression change of the *Ehrho* genes in *EhracM* gene silenced (gs) strain.** The average transcript amount of *Ehrho* genes based on three biological replicates. The 6 rows on the left indicate the RNA expression level of each sample. 1st, 2nd, and 3rd indicate three biologically independent replicates. The *EhracM* gs Average column indicates the transcript amount of each gene in the *EhracM* gs strain, whereas the psAP2 Average column shows that of the psAP2 mock strain, respectively, based on three biological replicates. The expression level of *EhracM* is highlighted in orange.
(XLSX)

**S8 Table. Expression change of the *Ehrho* genes in *EhracJ* gene silenced (gs) strain.** The average transcript amount of *Ehrho* genes based on three biological replicates. The 6 rows on the left indicate the RNA expression level of each sample. 1st, 2nd, and 3rd indicate three biologically independent replicates. The *EhracJ* gs Average column indicates the transcript amount of each gene in the *EhracJ* gs strain, whereas the psAP2 Average column shows that of the psAP2 mock strain, respectively, based on three biological replicates. The expression level

of *EhracJ* is highlighted in orange.
(XLSX)

**S9 Table. List of the hits in HA-EhRacM co-IP (less strict version).** The hits, which showed more than two-fold higher quantitative value (QV) in HA-EhRacM than in mock strain at least once in the three independent co-IPs, are listed. Each value indicates QV except for the "QV ratio" column. The hits exclusively identified in HA-EhRacM are surrounded by a bold-line box. The hits are listed in order of QV ratio. The hits related to Rho signaling are highlighted in orange, uridine/cytidine kinase are highlighted in pink, those related to proteasome systems are highlighted in green, and those related to ribosomes are highlighted in dark green.
(XLSX)

**S10 Table. List of the hits in HA-EhRacJ co-IP.** The hits in a co-IP, which showed more than two-fold higher quantitative value (QV) in HA-EhRacJ than in the mock strain, are listed. Each value indicates QV except for the "QV ratio" column. The hits exclusively identified in HA-EhRacJ are surrounded by a bold-line box. The hits are listed in order of QV ratio. Twenty-seven hits involved in the cytoskeleton binding or regulation are highlighted in pink, which were detected exclusively in HA-EhRacJ with quantification values (QV) greater than 5 or more than three times higher in HA-EhRacJ compared to the mock control.
(XLSX)

**S1 Video. Motility assay.** Representative movies of the live amoeba trophozoites (magenta) migrating on a glass surface. (A): psAP2 mock control strain, (B): *EhracM* gene silenced strain. The videos were acquired with CQ1. Each cell's track was visualized by each line.
(MP4)

**S2 Video. Live imaging of GFP-EhRacM overexpressing cells.** GFP-EhRacM overexpressing trophozoites were incubated in BIS medium and observed under spinning disk confocal microscope system.
(MP4)

**S3 Video. Recruitment of GFP-EhRacM to a macropinosome.** GFP-EhRacM overexpressing trophozoites were incubated in RITC dextran-containing BIS medium to observe the involvement of GFP-EhRacM in macropinocytosis. The white arrowheads show the representative macropinosome formation. (A) RITC dextran (red) and GFP-EhRacM are visualized. (B) Only GFP-EhRacM is visualized.
(MP4)

**S4 Video. Recruitment of GFP-EhRacM to a macropinosome after the removal of the F-actin envelope.** GFP-EhRacM overexpressing trophozoites were stained by SiR-Actin and incubated in RITC dextran-containing BIS medium to observe the timing of F-actin envelope removal and GFP-EhRacM recruitment to the macropinosomes. The white arrowheads show the representative macropinosome formation. (A) RITC dextran (red) and SiR-Actin (magenta) are visualized. (B) GFP-EhRacM is visualized.
(MP4)

**S1 Data. Excel spreadsheet containing, in separate sheets, the underlying numerical data and statistical analysis for Figs 1B, 2A, 2B, 3B, 4C, 5, 8, 9A, 9B and S2, S3A, S3B, S3C, S3D, S4, S5A, S5B, S8B, S10, S12, S14A and S14B.**
(XLSX)

## Acknowledgments

We thank all the members of the Nozaki laboratory at the University of Tokyo for the helpful discussion, Kumiko Shibata and Mihoko Imada for constructing several plasmids and providing technical support, and William Petri for the anti-Hgl antibody.

## Author Contributions

**Conceptualization:** Kumiko Nakada-Tsukui, Tomoyoshi Nozaki.

**Data curation:** Misato Shimoyama, Kumiko Nakada-Tsukui.

**Formal analysis:** Misato Shimoyama, Kumiko Nakada-Tsukui.

**Funding acquisition:** Misato Shimoyama, Kumiko Nakada-Tsukui, Tomoyoshi Nozaki.

**Investigation:** Misato Shimoyama, Kumiko Nakada-Tsukui.

**Methodology:** Misato Shimoyama, Kumiko Nakada-Tsukui, Tomoyoshi Nozaki.

**Project administration:** Kumiko Nakada-Tsukui, Tomoyoshi Nozaki.

**Resources:** Misato Shimoyama, Kumiko Nakada-Tsukui, Tomoyoshi Nozaki.

**Supervision:** Tomoyoshi Nozaki.

**Validation:** Misato Shimoyama, Kumiko Nakada-Tsukui.

**Visualization:** Misato Shimoyama.

**Writing – original draft:** Misato Shimoyama.

**Writing – review & editing:** Kumiko Nakada-Tsukui, Tomoyoshi Nozaki.

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
