## [Decision Letter · Decision Letter 0]

29 Jul 2024

Dear Prof. Nozaki,

Thank you very much for submitting your manuscript "EhRacM differentially regulates macropinocytosis and motility in the enteric protozoan parasite *Entamoeba histolytica*" for consideration at PLOS Pathogens. As with all papers reviewed by the journal, your manuscript was reviewed by members of the editorial board and by several independent reviewers. The reviewers appreciated the attention to an important topic. Based on the reviews, we are likely to accept this manuscript for publication, providing that you modify the manuscript according to the review recommendations.

The suggestions of reviewers 1 and 2 should be addressed (data presentation and various aspects of the text of the manuscript), but the new experimentation suggested by reviewer 1 is not needed.

Sincerely,

Katherine S Ralston, PhD

Guest Editor

PLOS Pathogens

Margaret Phillips

Section Editor

PLOS Pathogens

Michael Malim

Editor-in-Chief

PLOS Pathogens

orcid.org/0000-0002-7699-2064

The suggestions of reviewers 1 and 2 should be addressed (data presentation and various aspects of the text of the manuscript), but the new experimentation suggested by reviewer 1 is not needed.

Reviewer Comments (if any, and for reference):

Reviewer's Responses to Questions

**Part I - Summary**

Reviewer #1: This study by Shimoyama et al. examines the role of Rac GTPases in micropinocytosis and motility of Entamoeba histolytica. This parasite is the causative agent of amoebic dysentery and amoebic liver abscess. Since endocytosis and motility are important virulence determinants, the works provides important insight into pathogenicity. The work will be of interest to the Entamoeba community of researchers. Strengths of the study include the rigor and thoroughness, excellent use of controls, and excellent use of sophisticated molecular biology techniques and image analysis.

Reviewer #2: The manuscript entitled,” EhRacM differentially regulates macropinocytosis and motility in the enteric protozoan parasite Entamoeba histolytica” nicely shows involvement of another Rho like molecule in specifically in macropinocytosis. As E. histolytrica displays variety of endocytic processes, which are important for its survival and also impart its ability to invade host gut system, it becomes important to understand these processes in detail to develop anti amebic agents. The study harnesses the potential of gene silencing experiments to show the involvement of gene in macropinocytosis. Further, clear microscopic imaging and observation has led to place the recruitment of EhRacM in vesicular trafficking post actin coat disassembly. Although, involvement in phagocytosis cannot be ruled out as anything above 0.5µm is taken by phagocytosis. However, the study is very important to fill gaps in the signaling pathways which are not well explored in E. histolytica. The combination of different strategies to identify involvement of a gene in an endocytic process is beatifully executed. Addition of few explanation and small experiments may improve its overall content for readers.

Reviewer #3: This paper reports the discovery of how macropinocytosis and motility are regulated in the protozoan parasite Entamoeba histolytica. Both of these processes are central to the biology and pathogenesis of this extracellular gut-dwelling parasite. The authors started with gene silencing of the most highly expressed race and rho family proteins, identifying EhracM as a negative regulator of dextran phagocytosis. This discovery was validated by gene over-expression and identification of the sub cellular localization of EhracM. the work is rigorously performed and clearly presented and represents a significant advance in the understanding of E. histolytica phagocytic mechanisms.

**Part II – Major Issues: Key Experiments Required for Acceptance**

Reviewer #1: Potentially, one of the most important findings is that EhRacM accumulates on macropinosomes after they are formed (i.e., EhRacM does not accumulate on forming cups). The data supporting this finding are first shown in Figure 5 and form the basis for many of the next experiments (Figs. 6-8). Although the accumulation of EhRacM around one of the macropinosomes (white arrowhead) is evident and supported by the intensity measurements, there are other RITC-filled macropinosomes in the image that don’t seem to have an accumulation of EhRacM. For example, there is one just below the endosome labeled with the arrowhead that doesn’t seem to be enriched in EhRacM. This reviewer understands that there are different planes of focus, but it raises the question about whether the EhRacM-enriched endosome is an exception. It would be helpful to quantify the % of total macropinosomes that become enriched with EhRacM.

Reviewer #2: NA

Reviewer #3: none

**Part III – Minor Issues: Editorial and Data Presentation Modifications**

Reviewer #1: 1. In the introduction (Line 62), it sounds like viruses, bacteria and prions also exhibit their own micropinocytosis. This reviewer knows it is not what the authors mean (they mean that some of these pathogens are taken up into host cells by host cell endocytosis, but it is not clear as written.

2. In Line 123, the authors should include the gene name (rho1B) in addition to XP_651936. In this way, readers won’t need to backtrack to figure out what rac/rho could not be silenced.

Reviewer #2: 1. The author have introduced EhRacM as Rho like molecule but the sequence alignment with other known Rho like EhEhRho1 and host Rho proteins should be done to explain the relatedness and status of various domains and motifs in EhRacM. Incorporation of Phylogenetic analysis will be better to explain if it belongs to Rho class of GTPase.

2. Authors have used Verapamil to stain actin by SiR probe (line 818) in fixed trophozoites. However, verapamil is used for achieving higher labelling intensities in live cells during real time imaging. Can author explain why Verapamil was used for fixed cells?

3. According to figure 3A, the results indicate that EhRacM is involved in linear motility of trophozoites. However, the results are performed in normal conditions and random motility event is selected. If the motility assay can be performed in presence of some chemotactic agent like TGF-beta or cAMP gradient, then the comment on linear motility can be more affirmative.

4. Authors state that EhRacM is associated with macropinocytic vesicles for a long time and help in maturation (Fig 6). As this protein is also involved in phagocytosis and trogocytosis. Has it been reported in phagosome proteomes of E. histolytica reported by various authors?

5. What is the status of expression of this gene in virulent versus avirulent strains? Does virulent pathogen show decreased expression of this gene? Does treatment of trophozoites with Rac-GTPase inhibitors or constitutive expression of catalytically nonfunctional EhRacM mimics same effect as gene silenced one? Perform at least one of the experiments to validate gene silencing function.

6. Schematic representation of the model put forward by the authors in discussion part will be better for understanding by the readers.

Reviewer #3: none

PLOS authors have the option to publish the peer review history of their article (what does this mean?). If published, this will include your full peer review and any attached files.

Reviewer #1: No

Reviewer #2: No

Reviewer #3: **Yes: **William A. Petri, Jr.

Figure Files:

Data Requirements:

Reproducibility:

References:

---

## [Editor Report · Decision Letter 1]

23 Sep 2024

Dear Prof. Nozaki,

We are pleased to inform you that your manuscript 'EhRacM differentially regulates macropinocytosis and motility in the enteric protozoan parasite *Entamoeba histolytica*' has been provisionally accepted for publication in PLOS Pathogens.

Best regards,

Katherine S Ralston, PhD

Guest Editor

PLOS Pathogens

Margaret Phillips

Section Editor

PLOS Pathogens

Michael Malim

Editor-in-Chief

PLOS Pathogens

orcid.org/0000-0002-7699-2064
---

## [Editor Report · Acceptance letter]

9 Oct 2024

Dear Prof. Nozaki,

We are delighted to inform you that your manuscript, "EhRacM differentially regulates macropinocytosis and motility in the enteric protozoan parasite *Entamoeba histolytica*," has been formally accepted for publication in PLOS Pathogens.

Best regards,

Michael Malim

Editor-in-Chief

PLOS Pathogens

orcid.org/0000-0002-7699-2064